# Topoisomerase VI senses and exploits both DNA crossings and bends to facilitate strand passage

Timothy J Wendorff[1], James M Berger[2]*

[1]Biophysics Graduate Program, University of California, Berkeley, Berkeley, United States; [2]Department of Biophysics and Biophysical Chemistry, Johns Hopkins University School of Medicine, Baltimore, United States

**Abstract** Type II topoisomerases manage DNA supercoiling and aid chromosome segregation using a complex, ATP-dependent duplex strand passage mechanism. Type IIB topoisomerases and their homologs support both archaeal/plant viability and meiotic recombination. Topo VI, a prototypical type IIB topoisomerase, comprises two Top6A and two Top6B protomers; how these subunits cooperate to engage two DNA segments and link ATP turnover to DNA transport is poorly understood. Using multiple biochemical approaches, we show that Top6B, which harbors the ATPase activity of topo VI, recognizes and exploits the DNA crossings present in supercoiled DNA to stimulate subunit dimerization by ATP. Top6B self-association in turn induces extensive DNA bending, which is needed to support duplex cleavage by Top6A. Our observations explain how topo VI tightly coordinates DNA crossover recognition and ATP binding with strand scission, providing useful insights into the operation of type IIB topoisomerases and related meiotic recombination and GHKL ATPase machineries.

DOI: https://doi.org/10.7554/eLife.31724.001

## Introduction

The appropriate control of transcription, DNA replication, and chromosome segregation are essential to cell proliferation. These three processes are antagonized, however, by the double helical structure of DNA, which supercoils in response to helicase and polymerase activity and which promotes chromosome interlinkage during replicative synthesis (reviewed in [*Vos et al., 2011*; *Wang, 2002*]). In all cells, enzymes known as topoisomerases are used to overcome the natural topological impediments arising from these physical transactions that impinge on DNA. Many different topoisomerase families exist to relieve super-helical tension and DNA entanglements, all of which transiently form either single strand or double strand breaks to manipulate DNA topology (reviewed in [*Chen et al., 2013*; *Forterre and Gadelle, 2009*; *Schoeffler and Berger, 2008*]).

Type II topoisomerases introduce transient double strand breaks into DNA, and play a key role in unlinking catenated DNA molecules (*Holm et al., 1985*; *Spell and Holm, 1994*; *Zechiedrich and Cozzarelli, 1995*). The type IIA subfamily of topoisomerases, which are principally used in bacteria and eukaryotes, utilize a so-called 'two-gate' mechanism (*Roca et al., 1996*; *Roca and Wang, 1994*; *Roca and Wang, 1992*), in which one DNA duplex (termed the transport- or 'T'-segment) is captured by one half of the enzyme, actively passed through a second, protein-bound DNA duplex (the gate- or 'G'-segment), and expelled through the other end of the enzyme. T-segment capture is regulated by the ATP-dependent closure of one subunit dimer interface (*Roca and Wang, 1992*; *Wigley et al., 1991*), referred to as the 'ATP-gate', while a pair of catalytic tyrosine residues responsible for G-segment cleavage and opening reside in a second, separable subunit-subunit contact point termed the 'DNA-gate' (*Berger et al., 1996*; *Morais Cabral et al., 1997*; *Morrison and*

*For correspondence:
jberge29@jhmi.edu

**eLife digest** Each human cell contains genetic information stored on approximately two meters of DNA. Like holiday lights in a storage box, packing so much DNA into such a small space leads to its entanglement. This snarled DNA prevents the cell from properly accessing and copying its genes.

Type II topoisomerases are a group of enzymes that remove DNA tangles. They attach to one segment of a DNA tangle, cut it in half, remove the knot, and then repair the broken DNA strand. The process requires the proteins to 'burn' chemical energy. If topoisomerases make mistakes when they cut and reseal DNA, they could damage genetic information and harm cells. It is still unclear how these proteins recognize DNA tangles and use energy to remove knots instead of adding them.

Here, Wendorff and Berger use biochemical approaches to look into topo VI, a type II topoisomerase found in plants and certain single-celled organisms. When DNA is tangled, it forms sharp bends and crossings. Their experiments reveal that topo VI has certain 'sensors' that detect where DNA bends, and others that recognize the crossings. Only when both features are present does the enzyme start working and using energy. These sensors act as fail-safes to ensure that topo VI only breaks DNA when it encounters a proper knot, and is not 'set loose' on untangled DNA.

Future work will look at topo VI at an atom-by-atom level to reveal how exactly the enzymes 'see' DNA bends and crossings, and how interactions with the correct type of DNA triggers energy use and DNA untangling. Knowing more about topo VI can help researchers to understand how human and bacterial topoisomerases work. These results could also be generalized to other enzymes, for example those that help the genetic processes at play when sperm and egg cells form.
DOI: https://doi.org/10.7554/eLife.31724.002

*Cozzarelli, 1979*; *Tse et al., 1980*). In most instances, repeated cycles of ATP binding and hydrolysis allow for the processive removal of multiple DNA crossings. The use of ATP by type II topoisomerases in general has been proposed to serve as a mechanism for preventing the inappropriate formation of potentially cytotoxic DNA breaks (*Bates et al., 2011*); however, the molecular basis for the coupling between ATP turnover and DNA cleavage has remained enigmatic for the superfamily as a whole.

The type IIB topoisomerases, which are exemplified by DNA topoisomerase VI (topo VI) (*Bergerat et al., 1997*; *Bergerat et al., 1994*; *Forterre et al., 2007*), share evolutionarily conserved catalytic elements with their type IIA counterparts but are structurally distinct (*Corbett and Berger, 2003*; *Nichols et al., 1999*). Topo VI comprises an $A_2B_2$ heterotetramer formed by two Top6A and two Top6B subunits: Top6A forms a 'U'-shaped dimer that serves as the DNA-gate for G-segment cleavage and opening (*Bergerat et al., 1997*; *Nichols et al., 1999*), while Top6B constitutes the ATP-gate and dimerizes in response to nucleotide binding (*Corbett and Berger, 2003*). Topo VI is thought to serve as the primary topoisomerase for DNA decatenation and supercoil relaxation in archaea and is required for endoreduplication and cell growth in plants (*Bergerat et al., 1997*; *Bergerat et al., 1994*; *Hartung et al., 2002*; *Sugimoto-Shirasu et al., 2002*; *Yin et al., 2002*). Topo VI is also found sporadically throughout the bacterial domain, and a single chain variant, topo VIII, is found in certain plasmid-based mobile elements as well (*Forterre et al., 2007*; *Gadelle et al., 2014*). Interestingly, the type IIB topoisomerase scaffold has been co-opted to serve as the machinery responsible for introducing double-strand DNA breaks to initiate meiotic recombination in eukaryotes (*Bergerat et al., 1997*; *Keeney et al., 1997*; *Robert et al., 2016*; *Vrielynck et al., 2016*). How topo VI and its cousins engage DNA segments has yet to be determined.

The ATPase region is generally well-preserved between type IIA and type IIB topoisomerases, with the exception of an additional helix-two-turn-helix (H2TH) domain of unknown function found in topo VI and topo VIII (*Bergerat et al., 1997*; *Corbett and Berger, 2003*; *Gadelle et al., 2014*; *Wigley et al., 1991*). By contrast, the catalytic domains that comprise the DNA breakage-reunion region of type IIA and IIB enzymes have been extensively shuffled. One consequence of this rearrangement is that Top6A lacks a third subunit-subunit interface present in the type IIA enzymes, the 'C-gate' dimerization domain, which is thought to help mitigate the risk of aberrant double-strand break formation (*Bates et al., 2011*; *Berger et al., 1996*; *Nichols et al., 1999*; *Roca, 2004*). To compensate for the loss of this element, type IIB topoisomerases appear to have evolved a stringent

mechanism for controlling strand scission by Top6A that represses transesterase activity until ATP productively binds to Top6B (*Buhler et al., 1998*; *Buhler et al., 2001*). How Top6B activates Top6A is unknown; however, given that Spo11, a paralog of Top6A used in meiotic recombination, is also thought to require activation for DNA cleavage, aspects of this control mechanism may be broadly conserved (*Lam and Keeney, 2014*).

To better understand how type IIB topoisomerases coordinate DNA cleavage, we performed a comprehensive biochemical investigation of *Methanosarcina mazei* topo VI, a model mesophilic type IIB topoisomerase. We find that topo VI discriminates between linear and supercoiled DNA using an extensive and unanticipated DNA binding interface that specifically recognizes DNA crossings. Both gate closure and ATP hydrolysis by Top6B as well as transesterase activity by Top6A require engagement along this entire interface. Site-directed mutagenesis studies show that three conserved, positively charged regions on Top6B sense both the DNA bends and crossings present in supercoiled substrates and further serve to couple the binding of DNA crossings to B-subunit dimerization, nucleotide turnover, and DNA strand scission. Our results explain why type IIB topoisomerases absolutely depend upon the ATPase activity of the B-subunit to generate double strand breaks. These observations in turn reinforce the functional importance for DNA bending and potential T-segment-sensing elements in the related type IIA topoisomerases, and also provide insights as to how recently discovered meiotic Top6B homologs might promote Spo11 mediated strand scission during meiotic recombination.

## Results

### Topo VI is a distributive DNA relaxase that preferentially recognizes DNA crossings

We began our investigations of type IIB topoisomerase mechanism by measuring the affinity of *M. mazei* topo VI for DNAs of varying length or topological status. The relative affinity of the holoenzyme for fluorescein-labeled duplex DNAs ranging from 20 bp to 70 bp in length was assessed using a fluorescence anisotropy-based approach (the predicted G-segment binding channel of a Top6A dimer is ~16–20 bp in length [*Nichols et al., 1999*]). The DNA sequence used for these oligomers was based on a previously determined cleavage hotspot for *Sulfolobus shibatae* topo VI (*Buhler et al., 2001*) (*Figure 1—source data 1*). These experiments showed that whereas a 20 bp duplex binds relatively weakly to topo VI, apparent affinity increases with length, plateauing between 40 and 70 bp (*Figure 1A*, *Figure 1—source datas 2–3*). As the binding isotherms did not show any sign of complex interactions (such as cooperativity) and could be fit well by a single-site binding model (*Heyduk and Lee, 1990*), this result provided the first clue that topo VI might have more extensive interactions with DNA than previously hypothesized.

To determine whether topo VI displayed any preference for the topological status of DNA, the relative binding affinities of the holoenzyme were next assessed for supercoiled plasmid vs. sheared, linear salmon-sperm DNA using a competitive binding assay. Topo VI was incubated with the fluorescein-labeled 70 bp duplex DNA and varying amounts of unlabeled, supercoiled plasmid or sheared salmon-sperm DNA. The relative affinity of topo VI for each substrate was determined by monitoring how well the competitor DNAs interfered with binding of the labeled probe. The response to the titration of supercoiled DNA or sheared salmon-sperm DNA was fit to an explicit competitive binding model (*Wang, 1995*) to indirectly estimate affinities for the unlabeled substrates. Based on these measurements, topo VI showed a ~60-fold preference for supercoiled DNA ($K_{I,app} = 0.6 \pm 0.3$ nM) compared to sheared salmon-sperm DNA ($K_{I,app} = 39.3 \pm 2.6$ nM) (*Figure 1B*).

The difference in affinity between supercoiled and linear DNA suggested that topo VI might preferentially engage supercoiled substrates by binding to DNA crossings, DNA bends, or both. To distinguish between these modes, we examined the time-dependent processivity of topo VI in relaxing negatively supercoiled DNA. For type II topoisomerases in general, processivity describes the ability of a single enzyme to remain bound to a G-segment DNA during multiple strand passage events. For topo VI, the progress of ATP-dependent supercoil relaxation was followed by native agarose-gel electrophoresis, using a slight molar excess of plasmid over enzyme to disfavor the binding of two topo VI molecules to a single DNA substrate (*Figure 1C*). A highly processive topoisomerase, such as *Saccharomyces cerevisiae* topoisomerase II (*Sc*Top2), removes the majority of supercoils on a

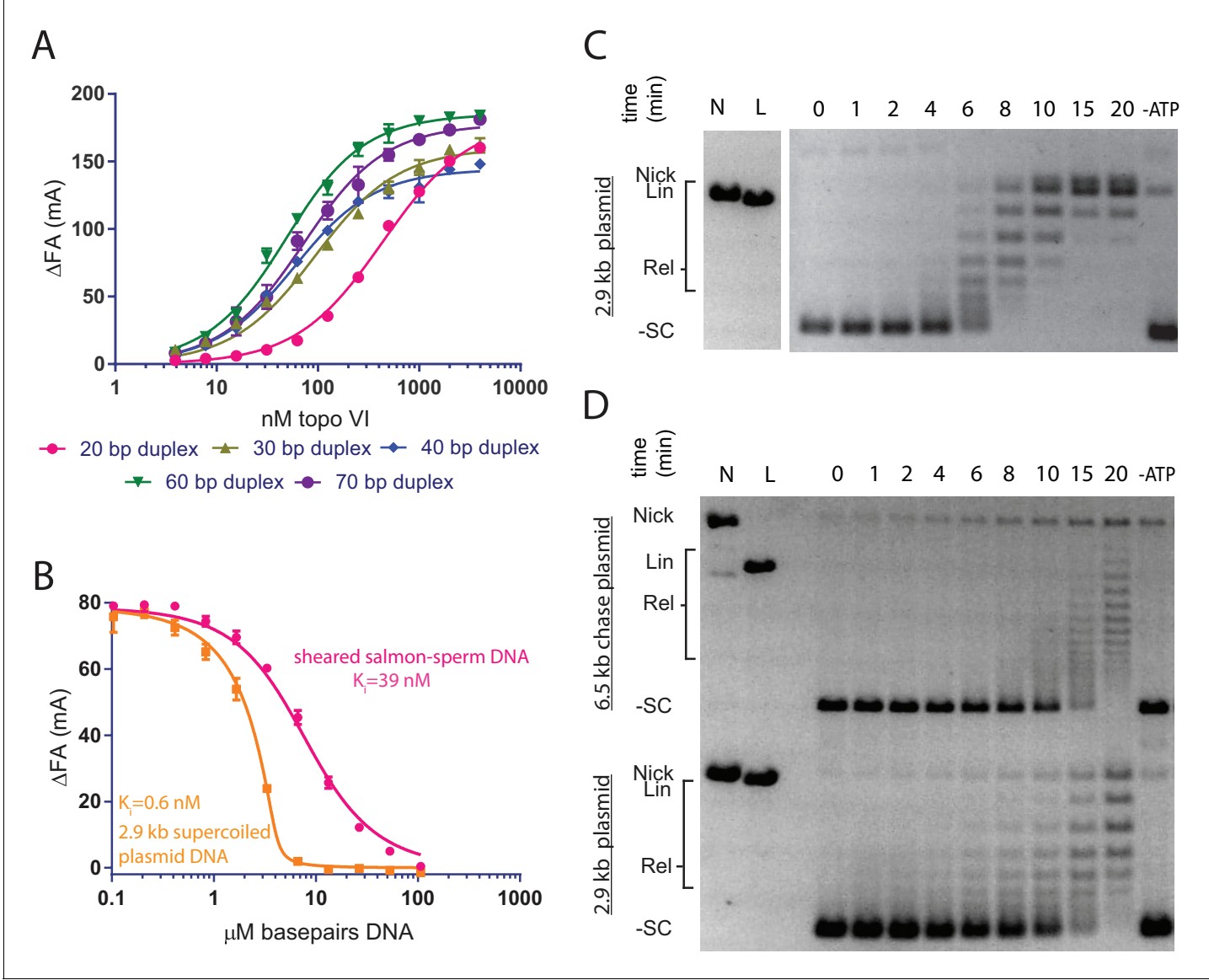

**Figure 1.** Topo VI binds longer duplexes and preferentially engages features of supercoiled DNA. (**A**) Binding of a 20, 30, 40, 60 or 70 bp fluorescein-labeled duplex (20 nM, sequences in *Figure 1—source data 1*) to topo VI, observed as a change in fluorescence anisotropy (ΔFA) measured in milli-anisotropy units (mA) as a function of enzyme concentration. Points and error bars correspond to the mean and standard deviation of three independent experiments. Curves represent fits to a single-site ligand depletion binding model. Apparent dissociation constants are reported in *Figure 1—source data 2*. (**B**) Fluorescence anisotropy experiment assessing the ability of supercoiled DNA and sheared salmon-sperm DNA to compete a fluorescein-labeled 70 bp duplex (20 nM duplex, 1.4 μM bp) from topo VI (100 nM). Non-labeled DNA was titrated from 0.1 μM bp to 106.5 μM bp and competition was observed as a change in fluorescence anisotropy (ΔFA) as measured in milli-anisotropy units (mA). Data are plotted as a function of the base-pair concentration (μM) of competing DNA. Points and error bars correspond to the mean and standard deviation of three independent experiments. Curves represent a fit to a competitive displacement model. Numerical data for (**A–B**) are reported in *Figure 1—source data 3*. (**C–D**) Test of processive supercoil relaxation by topo VI on negatively supercoiled plasmid DNA. Topo VI was pre-incubated in a 1:1.4 ratio to a 2.9 kb negatively supercoiled plasmid (6.7 ng/μL in assay). Reactions were started by addition of either (**C**) ATP or (**D**) ATP and a 6.5 kb 'chase' plasmid (6.7 ng/μL in assay) to compete for unbound enzyme. Samples were quenched at 0, 1, 2, 4, 6, 8, 10, 15, and 20 min. Each condition was also incubated without ATP for 20 min as a negative control. Plasmid size and topoisomer species are indicated to the left of each gel. For an example of processive supercoil relaxation by a type II topoisomerase, see *Figure 1—figure supplement 1*.

DOI: https://doi.org/10.7554/eLife.31724.003

The following source data and figure supplement are available for figure 1:

**Source data 1.** Oligonucleotides used for fluorescence anisotropy and FRET experiments.
DOI: https://doi.org/10.7554/eLife.31724.005
**Source data 2.** Binding affinities of topo VI for different length duplexes.

*Figure 1 continued on next page*

Figure 1 continued

DOI: https://doi.org/10.7554/eLife.31724.006

**Source data 3.** Numerical data associated with *Figure 1*.

DOI: https://doi.org/10.7554/eLife.31724.007

**Figure supplement 1.** Topo II processively relaxes supercoiled DNA as compared to topo VI.

DOI: https://doi.org/10.7554/eLife.31724.004

closed circular DNA in a single enzyme-DNA encounter (*Figure 1—figure supplement 1*) as evidenced by a paucity of intermediate DNA topoisomers between the supercoiled substrate and fully relaxed plasmid product. In contrast, topo VI produced a broad distribution of intermediate topoisomers that were gradually converted to the fully relaxed distribution, a behavior more consistent with low processivity.

To more thoroughly investigate supercoil processing by topo VI, we followed plasmid relaxation using two differently sized plasmids in a chase experiment. Following pre-incubation of a defined amount of topo VI with a slight molar excess of a primary 2.9 kb plasmid, a second, larger plasmid (6.5 kb) was added along with ATP to serve as a competing substrate for any dissociated enzymes (*Figure 1D*). In the case of a processive enzyme such as *Sc*Top2, the competing plasmid does not alter the initial rate at which a fully relaxed topoisomer distribution of the primary plasmid is generated (*Figure 1—figure supplement 1*). By contrast, topo VI again displayed clearly distributive behavior, relaxing both plasmids more slowly and simultaneously. Although assay conditions can modulate whether a topoisomerase acts processively or distributively (salt concentration in particular), both timecourse experiments were run under low-salt conditions where type IIA topoisomerases are primarily processive. Collectively, these findings demonstrate that topo VI operates by a principally distributive mechanism, whereby once a DNA crossing is resolved by strand passage, the enzyme will tend to dissociate from the substrate before acting on a new crossing and/or bent DNA segment.

## Topo VI actively uses DNA crossings to couple ATP hydrolysis with DNA strand passage

A defining characteristic of type II topoisomerases is the coupling of ATP turnover with efficient and rapid strand passage. In type IIA topoisomerases, DNA binding strongly stimulates ATPase activity (*Lee et al., 2013*; *Lindsley and Wang, 1993*; *Liu et al., 1979*; *Mizuuchi et al., 1978*; *Osheroff et al., 1983*; *Sugino and Cozzarelli, 1980*). However, the coupling of DNA topological state to the magnitude of the ATP hydrolysis stimulation varies between different type IIA homologs (*Anderson et al., 1998*; *Gubaev and Klostermeier, 2011*; *Harkins and Lindsley, 1998*; *McClendon et al., 2005*; *Osheroff et al., 1983*; *Sugino and Cozzarelli, 1980*; *Vaughn et al., 2005*). To determine whether the ATPase activity of type IIB topoisomerases is stimulated in a DNA topology-dependent manner, we examined nucleotide turnover by wild-type topo VI in the absence and presence of linear sheared salmon-sperm DNA and supercoiled plasmid DNA substrates at varying ATP concentrations using a coupled assay. Hydrolysis was also measured for an ATPase-deficient topo VI construct (Top6B$^{E44A}$) to identify non-specific activity arising from contaminating ATPases (*Figure 2A*, *Figure 2—figure supplement 1*). Although topo VI likely hydrolyzes ATP cooperatively, the data conformed to apparent Michalis-Menten behavior and were fit to this model (*Figure 2— source datas 1–2*). Topo VI showed negligible basal ATPase activity, and required the addition of a DNA substrate to hydrolyze ATP. Incubation with supercoiled DNA produced the maximal observed rate of hydrolysis (and decreased the $K_{m,app}$ for ATP), resulting in a ~5-fold increase in catalytic efficiency ($k_{cat,app}/K_{m,app}$) over that for sheared salmon-sperm DNA. The observation that supercoiled DNA is more effective than linear substrates in activating ATP turnover indicates that topo VI not only interrogates DNA for specific topological features, but that its activity is potentiated when such features are recognized.

ATP binding and hydrolysis by topo VI and many other enzymes that share its GHKL ATPase fold (e.g. type IIA topoisomerases, Hsp90, MutL, and MORC ATPases) rely on nucleotide-dependent dimerization of ATP-binding domains to elicit biological activity (*Ali et al., 2006*; *Ban et al., 1999*; *Ban and Yang, 1998*; *Dutta and Inouye, 2000*; *Li et al., 2016*; *Shiau et al., 2006*; *Wigley et al., 1991*). A mechanism in which supercoiled DNA binding, in particular T-segment engagement,

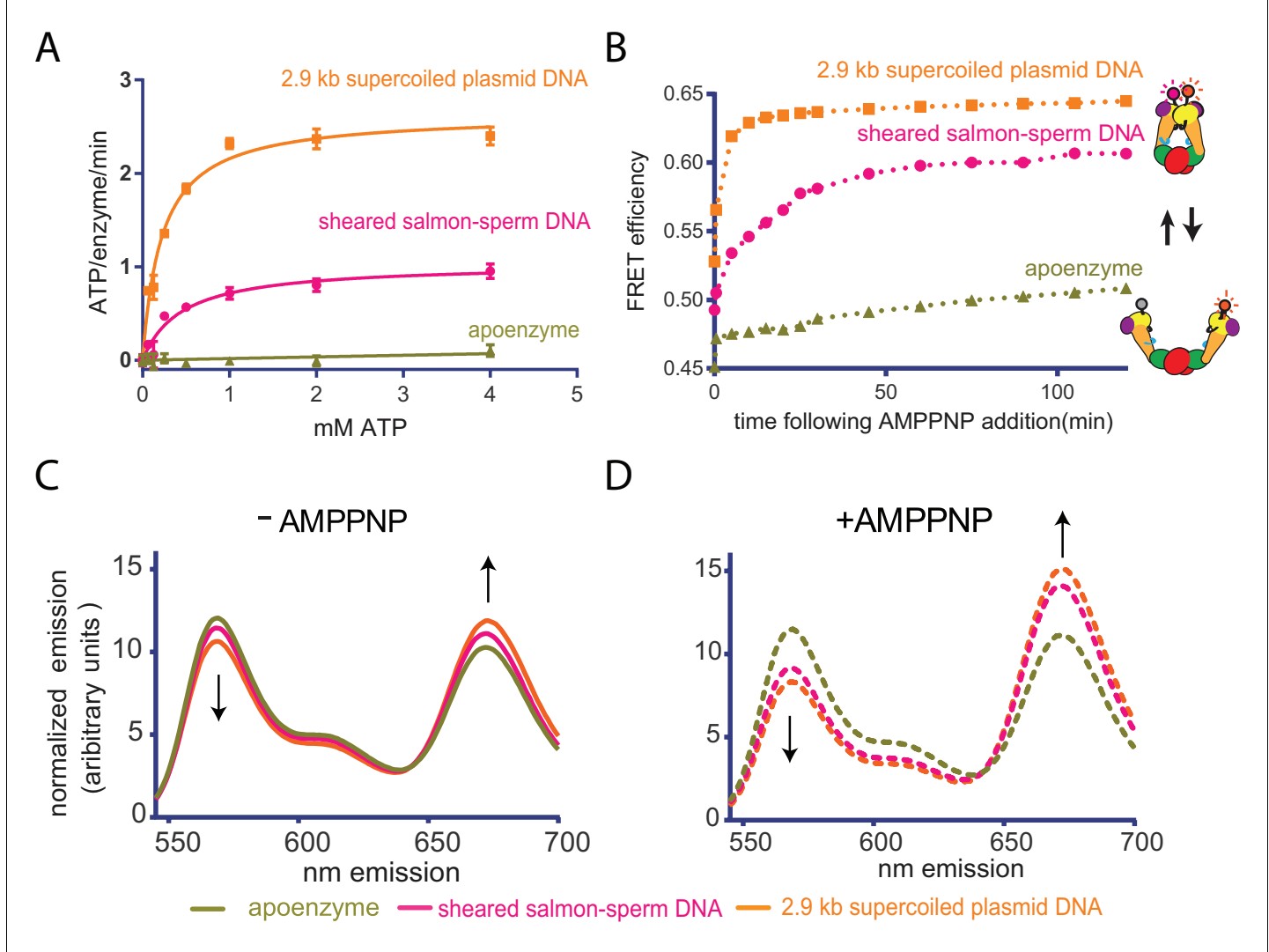

**Figure 2.** Top6B dimerization and ATP hydrolysis activity are stimulated by supercoiled DNA. (**A**) Rate of steady-state ATP hydrolysis catalyzed by topo VI alone, topo VI incubated with 400 μM basepairs of linear sheared salmon-sperm DNA (800:1 basepair to enzyme ratio), or topo VI incubated with 400 μM basepairs of 2.9 kb supercoiled plasmid DNA (800:1 basepair to enzyme ratio) as a function of ATP concentration. Rates were determined spectroscopically using an NADH-coupled assay. Data represent hydrolysis rates after subtracting a small contribution of non-specific ATPase activity from assays performed with an ATPase-deficient topo VI construct (**Figure 2—figure supplement 1**). Points and bars correspond to the mean and standard error of the mean of three independent experiments. Curves represent fits to a Michaelis-Menten kinetics model reported in **Figure 2— source data 1**. (**B**) The ratiometric FRET efficiency of Alexa555/Alexa647-labeled topo VI^cyslite-155C (see **Figure 2—figure supplement 2**) was monitored over time following the addition of AMPPNP for enzyme alone, enzyme bound to supercoiled DNA, or enzyme bound to short linear (sheared salmon-sperm) DNA. Numerical data for (**A–B**) are reported in **Figure 2—source data 2**. (**C–D**) Example fluorescence emission spectra produced by 530 nm excitation of Alexa555/Alexa647-labeled topo VI^cyslite-155C assessing the conformation of the Top6B ATPase domain in the absence of nucleotide ((**C**), solid lines) and 120 min following addition of AMPPNP ((**D**), dashed lines). Spectral emission was normalized by total emission from 545 nm to 700 nm. This behavior contrasts that of a model type IIA topoisomerase, ScTop2 (see **Figure 2—figure supplement 3**).

DOI: https://doi.org/10.7554/eLife.31724.008

The following source data and figure supplements are available for figure 2:

**Source data 1.** Apparent kinetic parameters for ATP hydrolysis by topo VI.
DOI: https://doi.org/10.7554/eLife.31724.012

**Source data 2.** Numerical data associated with **Figure 2**.
DOI: https://doi.org/10.7554/eLife.31724.013

**Figure supplement 1.** Determination of contaminating ATPase activity levels present in topo VI preparations using a hydrolysis-deficient mutant.
DOI: https://doi.org/10.7554/eLife.31724.009

**Figure supplement 2.** Design and production of a FRET pair-labeled topo VI to report on B-subunit conformation.

*Figure 2 continued*

DOI: https://doi.org/10.7554/eLife.31724.010

**Figure supplement 3.** AMPPNP is sufficient to close the ATP gate of *Sc*Top2.

DOI: https://doi.org/10.7554/eLife.31724.011

promotes Top6B dimerization could thus explain why supercoiled DNA stimulates ATP turnover. To test this idea, we developed a Förster Resonance Energy Transfer (FRET) assay to monitor ATPase domain dimerization in the context of the topo VI holoenzyme. We first identified and mutated surface cysteines to non-reactive residues to create a fully functional 'cys-lite' construct of the holoenzyme. Thr155 of Top6B was then substituted with cysteine (*Figure 2—figure supplement 2A*). Dual labeling with donor (Alexa 555-maleimide) and acceptor (Alexa 647-maleimide) fluorophores yielded an enzyme population containing an expected labeled mixture of correctly labeled donor-acceptor enzymes (50%), and both acceptor-acceptor (25%) and donor-donor (25%) labeled enzymes (*Figure 2—figure supplement 2B*; labeling efficiency was determined by spectral absorption). The labeled topo VI holoenzymes were able to fully relax DNA and showed only a slight impairment (~2 fold) of overall specific activity compared to wild-type topo VI (*Figure 2—figure supplement 2C*).

Using the labeled enzyme, bulk FRET efficiencies in the absence and presence of either linear or supercoiled DNA were first measured by scanning the spectral emission of both donor and acceptor fluorophores under excitation at 530 nm. The conformational response of the enzyme to AMPPNP, a non-hydrolyzable ATP analog, was then assessed for the enzyme alone and in the presence of each substrate over time (*Figure 2B*). The addition of sheared salmon-sperm DNA and to a greater extent supercoiled DNA, led to minor but reproducible increases in FRET efficiency (*Figures 2C*, 0 min time-point), suggesting that DNA binding alone alters the conformation of Top6B in the holoenzyme. By comparison, the addition of AMPPNP led to larger FRET responses, and FRET efficiency increased much more rapidly with supercoiled DNA compared to linear sheared salmon-sperm DNA. AMPPNP alone produced detectable but minor FRET changes when DNA was omitted, indicating that duplex binding is needed for ATPase domain dimerization (*Figure 2D*, 120 min time-point). In conjunction with the ATPase data, these observations show that – unlike type IIA topoisomerases, whose ATPase regions efficiently dimerize in the absence of DNA ([*Gubaev and Klostermeier, 2011*; *Roca and Wang, 1992*] and *Figure 2—figure supplement 3*) – topo VI utilizes the DNA geometries presented by supercoiled substrates to help favor nucleotide-dependent conformational changes associated with strand passage.

## Three conserved elements in Top6B play a role in DNA binding, the sensing of DNA geometry, and the productive coupling of ATP hydrolysis to strand passage

Based on the ability of topo VI to recognize and utilize topological features in supercoiled DNA to promote activity, we set out to identify the structural elements responsible for this coupling. Working from an assumption that topology-sensing elements might consist in part of positively charged residues on the B subunit, we mapped both amino acid conservation (derived from a multiple sequence alignment of Top6B homologs) and electrostatic surface potential onto the known structure of *M. mazei* Top6B using ConSurf and ABPS (*Figure 3A–C* and (*Ashkenazy et al., 2010*; *Baker et al., 2001*)). By comparing positively charged interfaces against sequence conservation, we identified three different regions as candidate DNA interaction sites.

The first prospective locus consisted of a small loop of basic residues (KGRR$_{186-189}$) (*Figure 3D*) within the predicted T-segment storage cavity of topo VI. A second feature comprised a trio of conserved basic residues (R457, K399 and K401) that are found within two spatially adjacent structural elements (*Figure 3E*): the C-terminal, α-helical stalk of Top6B (which connects the so-called 'transducer' domain of this subunit to Top6A), and a loop in the transducer domain containing the so-called 'WKxY motif,' which is conserved in both Top6B and many meiotic Top6B-like proteins (*Robert et al., 2016*). The third area of note, the H2TH domain, is embedded between the topo VI GHKL and transducer regions. The function of the topo VI H2TH domain has not been established, but this type of fold serves as a general nucleic-acid-binding element in a diverse number of proteins, including FpG/Nei DNA glycosylases, s13 ribosomal proteins, and sIHF type nucleoid-

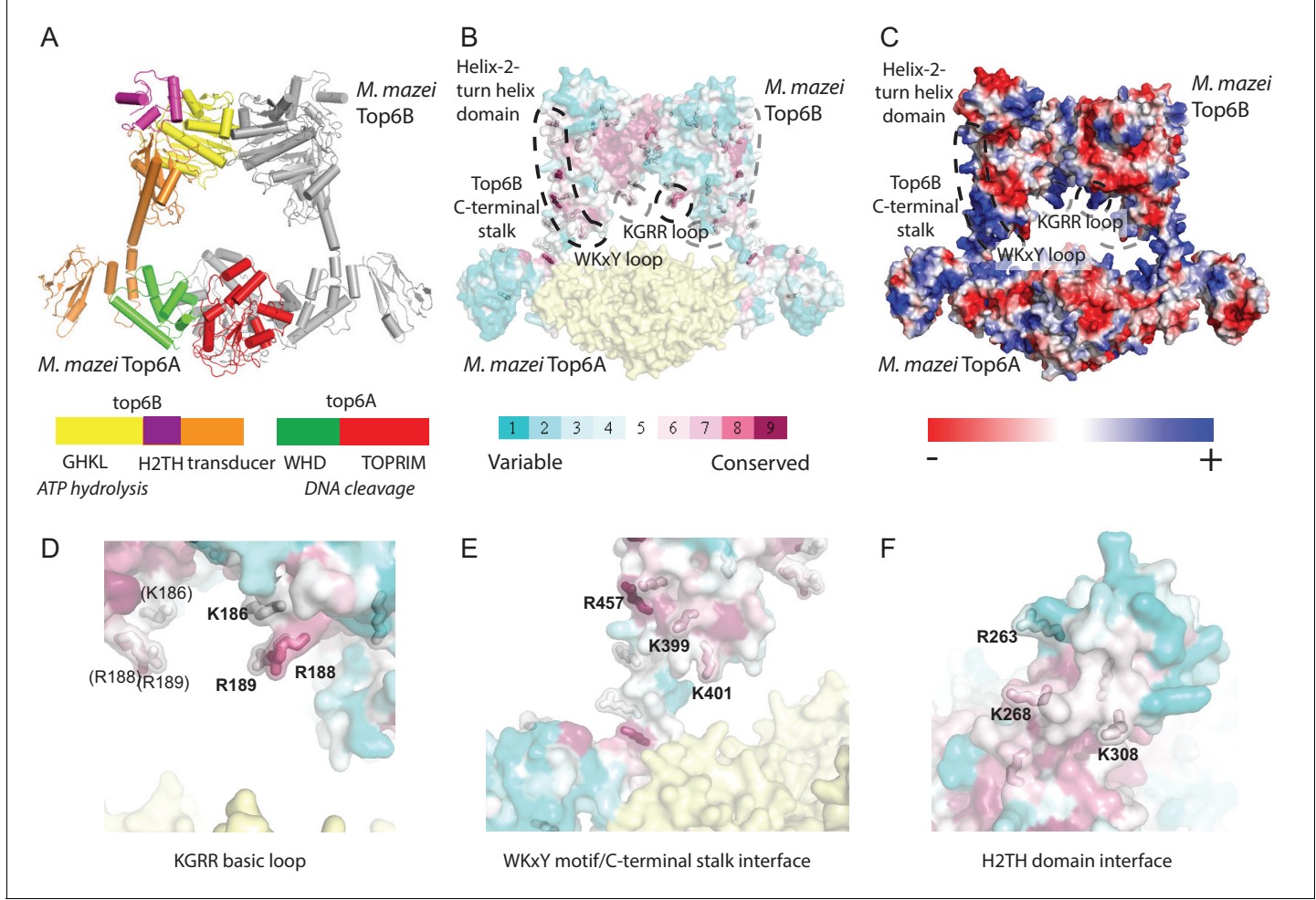

**Figure 3.** Identification of potential DNA-binding elements in Top6B. (A) Primary and tertiary structure [Protein Data Bank (PDB) ID: 2Q2E] of the *M. mazei* topo VI heterotetramer. Domains for one Top6A-Top6B heterodimer are colored as shown in the primary structure and the partner Top6A-Top6B heterodimer is shown in grey. Catalytic function is denoted in italics under primary structure. (B) Mapping of sequence conservation in Top6B based on a PSI-BLAST multiple sequence alignment. Conserved surface-exposed arginine and lysine residues (ConSurf score of $\geq$6) are shown as sticks. Coloration from cyan to magenta denotes variable to conserved. Top6A is represented in yellow. (C) Electrostatic surface representation of topo VI. A conserved basic loop in the T-segment storage cavity, and a conserved basic interface stretching from the WKxY motif and C-terminal stalk of Top6B to the Helix-2-turn-helix (H2TH) domain are labeled. (D) View of the KGRR basic loop motif. (E) View of the C-terminal stalk/WKxY interface. (F) View of the H2TH DNA-binding interface, rotated 90° towards the point of view as compared to A–C. See *Figure 3—figure supplement 1* for a more detailed rationale for the functional importance of this interface. In (D–F), residues mutated to alanine or to glutamate for functional studies are labeled. Mutations to these interfaces produced well-behaved functional mutants (see *Figure 3—figure supplement 2*).

DOI: https://doi.org/10.7554/eLife.31724.014

The following figure supplements are available for figure 3:

**Figure supplement 1.** Comparison between H2TH domain homologs predicts a DNA-binding interface in Top6B.
DOI: https://doi.org/10.7554/eLife.31724.015

**Figure supplement 2.** Topo VI functional mutants show similar solution properties to wild-type enzyme as judged by gel filtration.
DOI: https://doi.org/10.7554/eLife.31724.016

associated proteins (*Brodersen et al., 2002*; *Sugahara et al., 2000*; *Swiercz et al., 2013*; *Zharkov et al., 2002*). Comparison of nucleic-acid-bound H2TH domain structures with Top6B (*Figure 3—figure supplement 1*) highlighted R263, K268 and K308 as candidate residues that might interact with DNA (*Figure 3F*).

Having identified three potential sites for supercoil sensing on the surface of Top6B, six constructs were generated to assess the functional attributes of each region. Selected constructs included triple-neutral and triple-acidic mutations to the basic storage-cavity loop (KGRR→AGAA

and EGEE, referred to as KGRR$^{AAA}$ and KGRR$^{EEE}$), the C-terminal stalk (Stalk/WKxY$^{AAA}$ and Stalk/WKxY$^{EEE}$), and the H2TH domain (H2TH$^{AAA}$ and H2TH$^{EEE}$). All six mutant topo VI holoenzymes were soluble upon expression, purified to homogeneity (as judged by SDS-PAGE), and appeared well-behaved based on gel-filtration chromatography profiles as compared to the wild-type enzyme (*Figure 3—figure supplement 2*).

To assess overall activity, we next looked at the supercoil relaxation activity of the mutant enzymes compared to wild-type topo VI as a function of enzyme concentration (*Figure 4A*). Both sets of KGRR and Stalk/WKxY mutants (neutral and acidic) proved completely unable to relax super-coiled substrate. By contrast, both sets of mutations to the H2TH region led to enzymes that were able to relax supercoiled DNA, but with ~20–30 fold lower efficiency than native topo VI. The activity profiles seen in enzyme titration assays were corroborated by timecourse assays at a fixed enzyme concentration (*Figure 4—figure supplement 1*). In some of these experiments, the open-circle (nicked) plasmid species increased over time; however, this increase was independent of both nucleotide and topo VI, and thus does not reflect an elevated nicking activity of the mutants. Collectively, these findings show that the KGRR loop and the Stalk/WKxY region are essential components for topo VI function, but that the H2TH domain, while important, is not strictly required for strand passage.

To further investigate the role of each DNA-binding interface in the topo VI reaction cycle, the stimulatory effect of sheared salmon-sperm DNA and supercoiled DNA upon ATP hydrolysis activity of the six mutants was compared to wild-type enzyme. ATP hydrolysis rates were again measured using a coupled assay; however, ATP was held at 2 mM for these experiments, while the concentration of DNA substrate was varied to characterize the stimulatory effects of each substrate on each enzyme (*Figure 4B*, *Figure 4—source datas 1–2*). No DNA-stimulated ATP turnover was observed for either of the Stalk/WKxY mutants. Interestingly, the H2TH$^{AAA}$ and H2TH$^{EEE}$ mutants, which exhibited large defects in strand passage, showed similar levels of ATP hydrolysis stimulation by both DNA substrates as compared to wildtype topo VI. Moreover, whereas no additional ATP turnover was observed for the KGRR$^{AAA}$ and KGRR$^{EEE}$ mutants on sheared salmon-sperm DNA, both variants showed an increased maximal rate of DNA-stimulated ATP hydrolysis compared to the wild-type enzyme on supercoiled DNA (albeit with a more weakly coupled response to DNA concentration than wildtype topo VI or the H2TH mutants as judged by $K_{stim,DNA}$). All six mutants exhibited basal hydrolysis rates similar to both wild-type topo VI and the ATPase-deficient Top6AB$^{E44A}$ construct (*Figure 4—figure supplement 2*), indicating that the DNA-stimulated responses of each topo VI mutant are directly attributable to the introduced alterations. Collectively, these data indicate that the abrogation of strand passage activity by the KGRR loop mutants stems in part from a loss of an essential DNA-sensing motif required to carry out strand passage. However, unlike the Stalk/WKxY mutants, the KGRR loop mutants retain some feature which allows supercoiled, but not short linear DNAs, to promote ATP hydrolysis. Mutations to the H2TH domain additionally appear to largely decouple strand passage from ATP hydrolysis, yet do not appreciably alter the DNA dependence of ATPase activity. This result implies a role for the H2TH domain in facilitating A- and B-subunit coordination to minimize futile cycling.

Since all three interfaces identified affect strand passage activity and its coupling to ATP turnover, we next tested whether the observed differences result directly from weakened binding to duplex DNA. Using fluorescence anisotropy, the affinity of each mutant was assessed for a range of duplex lengths (30, 40, 60, and 70 bp) found to exhibit moderate-to-tight binding to wild-type topo VI (*Figure 5A* and *Figure 5—source datas 1* and *3*). As with native topo VI, a single-site binding model adequately described the DNA-binding isotherms for the mutant panel; the one exception was the data for the H2TH$^{EEE}$ mutant, which fit better to a cooperative model. This result suggests that charge reversal in the H2TH region may alter how longer duplexes are bound by the enzyme—although the direct binding data suggest that H2TH$^{EEE}$ binds longer DNAs better than H2TH$^{AAA}$, both mutants display similar affinities for a 60 and 70 bp duplex in competitive binding experiments (*Figure 5—figure supplement 1*), indicating that differences in the fluorophore environment may underlie the higher $K_{d,app}$ values seen in the direct binding study. Both Stalk/WKxY mutants were compromised for DNA binding overall (as judged by the maximum observed changes in anisotropy), with the magnitude of the binding defects proving more severe for the acidic substitutions. This finding highlights the Stalk/WKxY region of Top6B as an important DNA-binding interface, a finding that helps explain both why the binding affinity of topo VI is higher for DNAs whose length exceeds

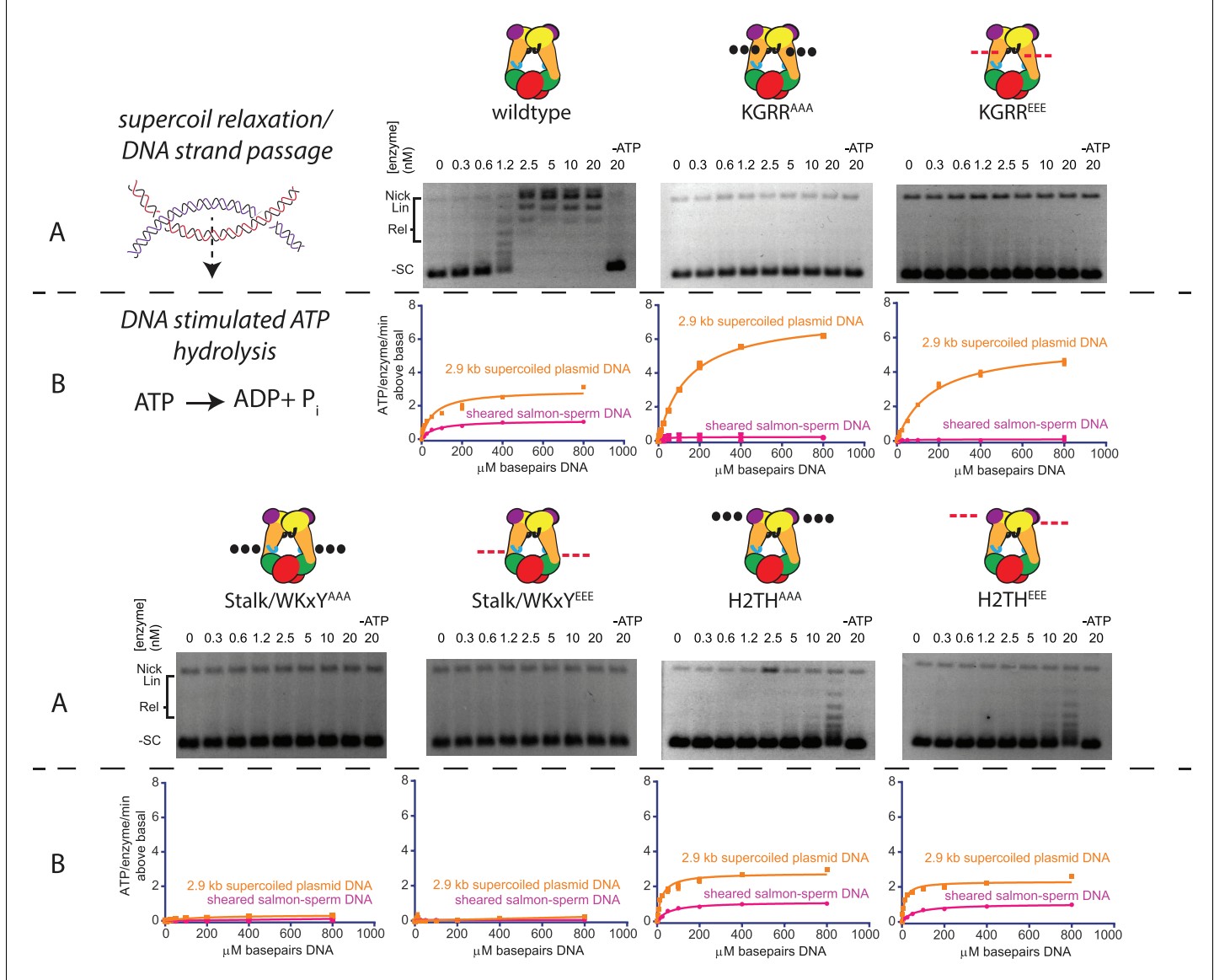

**Figure 4.** Effect of neutralization and charge reversal mutations to the KGRR loop, Stalk/WKxY region, or H2TH DNA-binding interface on supercoil relaxation activity and ATP hydrolysis by topo VI. (**A**) Activity of mutant topo VI constructs for relaxing supercoiled DNA compared to wild type as a function of enzyme concentration. For the enzyme titrations (0.3–20 nM in two-fold steps), each assay proceeded for 30 min prior to quenching with EDTA and SDS and contained 3.5 nM plasmid (10.2 µM bp DNA). Similar behavior for each mutant may be observed by timecourse (**Figure 4—figure supplement 1**). The placement and nature of mutations in each construct are depicted in the cartoons above each titration ('•••' - AAA; '- - -' – EEE). (**B**) The rate of steady-state ATP hydrolysis above basal levels (**Figure 4—figure supplement 2**) catalyzed by wild-type topo VI compared to mutant topo VI constructs, plotted as a function of the basepair concentration (µM) of sheared salmon-sperm DNA (pink), or a 2.9 kb supercoiled plasmid DNA (orange). ATP was held at 2 mM, and rates were determined spectroscopically using an NADH-coupled assay. Points and bars correspond to the mean and standard deviation of three independent experiments. Curves represent a fit to a Michealis-Menten type kinetics model reported in **Figure 4—source data 1**. Numerical data are reported in **Figure 4—source data 2**.

DOI: https://doi.org/10.7554/eLife.31724.017

The following source data and figure supplements are available for figure 4:

**Source data 1.** Kinetic parameters for DNA-dependent stimulation of topo VI ATPase activity.

DOI: https://doi.org/10.7554/eLife.31724.020

**Source data 2.** Numerical data associated with **Figure 4**.

DOI: https://doi.org/10.7554/eLife.31724.021

**Figure supplement 1.** Supercoil relaxation activity of topo VI mutants as a function of time.

DOI: https://doi.org/10.7554/eLife.31724.018

*Figure 4 continued on next page*

*Figure 4 continued*

**Figure supplement 2.** Comparison of basal ATPase activities for topo VI mutants.

DOI: https://doi.org/10.7554/eLife.31724.019

what is necessary to bind a Top6A dimer alone and why mutations in this region lead to defects in both strand passage and ATP hydrolysis. By comparison, the KGRR loop and the H2TH domain mutants showed either no change or only a moderate decrease (for the 60 and 70 bp duplexes) in DNA affinity compared to wildtype topo VI, suggesting that these regions potentially contribute a more peripheral or secondary site of DNA binding.

Because the KGRR loop and H2TH domain mutants minimally impacted affinity for short duplex DNAs as compared to the Stalk/WKxY mutants, we wondered whether these motifs might instead contribute to the preferential binding of topo VI seen for supercoiled DNA (*Figure 1B*). To this end, the relative affinities of supercoiled plasmid and linear, sheared salmon-sperm DNA were assessed for both sets of KGRR and H2TH mutants, using the fluorescence anisotropy-based competition assay described earlier (*Figure 5B*, and *Figure 5—source datas 2–3*). The H2TH$^{AAA}$ substitution minimally affected supercoiled DNA binding, whereas the H2TH$^{EEE}$ and both KGRR substitutions resulted in a ~10–20 fold decrease of the overall affinity of topo VI for supercoiled DNA, with KGRR$^{EEE}$ showing a greater defect than KGRR$^{AAA}$. Both KGRR substitutions adversely impacted the binding of random linear DNA compared to wild type as well, a result concordant with this mutant's negligible ATPase activity on sheared salmon-sperm DNA and which further suggests that this set of substitutions may ablate a secondary DNA-binding site on the holoenzyme. Together, these data indicate that both the KGRR loop and H2TH domain contribute to the preferential binding of topo VI to supercoiled substrates as compared to sheared salmon-sperm DNA, but that neither is solely responsible for this discrimination.

## The KGRR loop acts as a DNA crossing sensor to regulate Top6B dimerization

Rather than contributing to overall DNA affinity, the biochemical and biophysical activities of our topo VI mutants implicate the KGRR loop and H2TH domain in recognizing supercoiled DNA and in coupling ATP hydrolysis to strand passage. Although these two motifs might recognize either the DNA crossings or bends present in plectonemic substrates, we hypothesized that the KGRR element in particular might sense T-segment occupancy directly due to its physical location in the holoenzyme (*Figure 3*). To address this question, we designed a fluorescently-labeled, 20 bp by 16 bp Holliday junction substrate that can form a stacked-X structure (*Duckett et al., 1988*; *Ortiz-Lombardía et al., 1999*) as a mimic of a prospective duplex DNA crossing (*Figure 6A–B*). Using fluorescence anisotropy, wildtype topo VI was found to bind this substrate nearly 4-fold more tightly than a single 20 bp DNA duplex (*Figure 6C*, *Figure 6—source datas 1–2*). We next asked whether mutations to the KGRR loop or the H2TH domain interfered with binding to the stacked-junction substrate. Whereas both H2TH mutants showed similar increases in affinity for the stacked-junction DNA as seen with native topo VI, the KGRR$^{AAA}$ mutant showed a clear decrease in affinity for this substrate compared to a 20 bp duplex (and little to no change in affinity for a 20 bp duplex alone, *Figure 6C*, *Figure 6—source datas 1–2*). The KGRR$^{EEE}$ mutant displayed an even more pronounced defect in junction binding. Collectively, this response implicates the KGRR loop in the binding of DNA crossings by topo VI, potentially as a T-segment-sensing element.

We next considered whether the binding of DNA crossings facilitated by the KGRR loop might affect how supercoiled DNA promotes the ATP-dependent dimerization of Top6B (as observed for native topo VI [*Figure 2B*]), or whether this activity might instead arise from an H2TH domain interaction with supercoiled DNA. To address this question, we added the KGRR$^{AAA}$ and H2TH$^{AAA}$ mutations into the topo VI construct used to monitor the conformational status of the ATPase domain by FRET. Following purification and labeling, we measured the emission spectra of both mutants alone and bound to supercoiled DNA. Similar to the wild-type construct, both the KGRR$^{AAA}$ and H2TH$^{AAA}$ mutants showed increased FRET efficiencies in the presence of supercoiled substrate, independent of nucleotide (*Figure 6—figure supplement 1*). Interestingly, the 'enzyme alone' spectra suggest that each mutant alters the resting conformational status of the Top6B dimer compared to

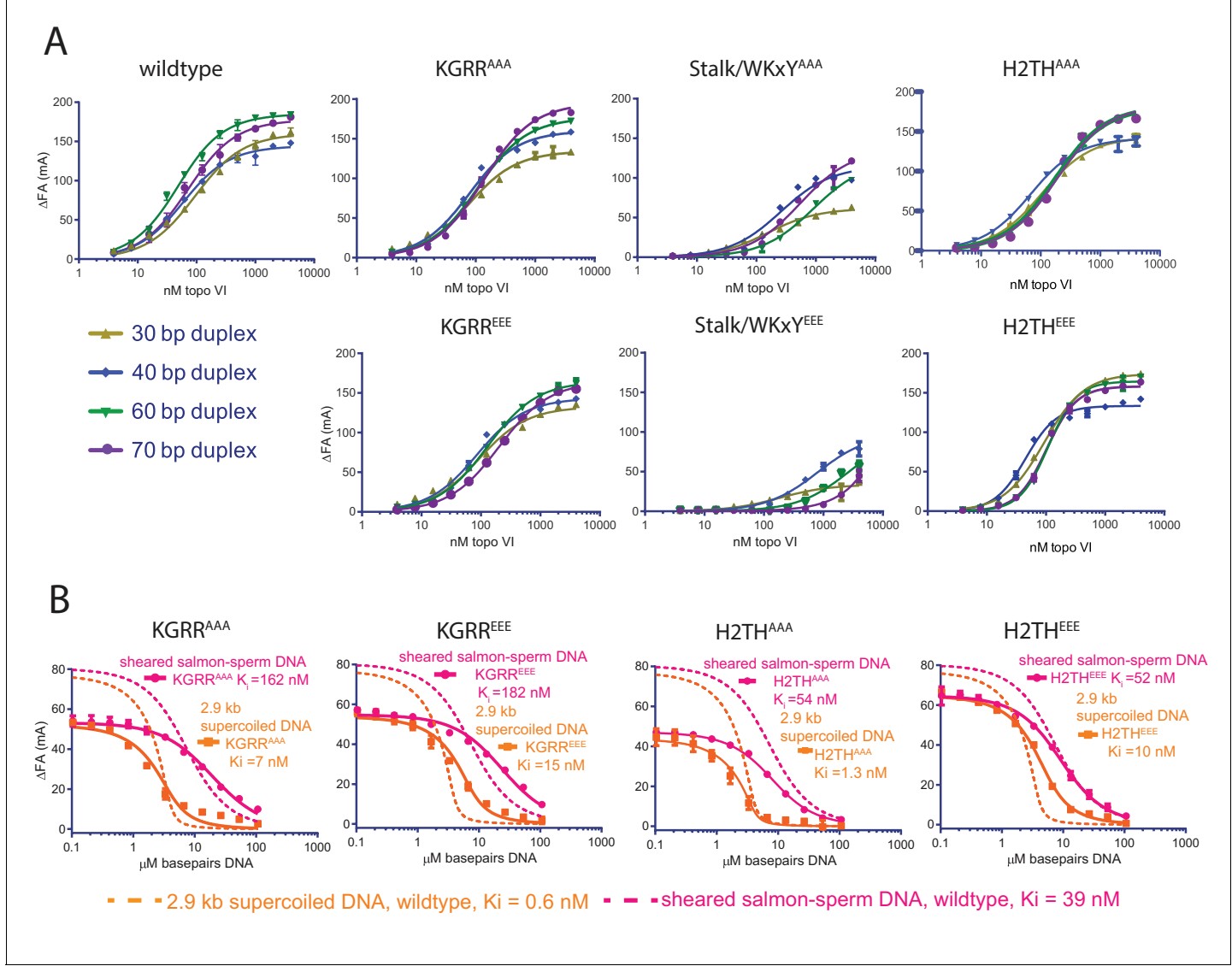

**Figure 5.** Effect of neutralization and charge reversal mutations to Top6B on DNA binding affinity and preferential engagement of supercoiled DNA. (**A**) Binding of a 30, 40, 60, or 70 bp fluorescein-labeled duplex (20 nM) to topo VI mutant constructs. Binding was observed as a change in fluorescence anisotropy (ΔFA) and measured in milli-anisotropy units (mA) as a function of enzyme concentration. Points and error bars correspond to the mean and standard deviation of three independent experiments. For the H2TH$^{EEE}$ mutant, curves represent fits to a Hill-type cooperative binding model. All other curves represent fits to a single site ligand depletion binding model. Binding isotherms for the wildtype enzyme are reproduced from *Figure 1A* for reference. Apparent dissociation constants are reported in *Figure 5—source data 1*. (**B**) Binding assay assessing the ability of supercoiled DNA and sheared salmon-sperm DNA to compete a fluorescein-labeled 70 bp duplex (20 nM duplex, 0.14 μM bp) from 100 nM H2TH$^{AAA}$, H2TH$^{EEE}$, KGRR$^{AAA}$ or KGRR$^{EEE}$, topo VI enzyme. Non-labeled DNA was titrated from 0.1 μM bp to 106.5 μM bp with competition observed as a change in fluorescence anisotropy (ΔFA) measured in milli-anisotropy units (mA). Data are plotted as a function of the basepair concentration (μM) of competitor DNA. Points and error bars correspond to the mean and standard deviation of three independent experiments. Curves represent a fit to an explicit competitive displacement model (*Figure 5—source data 2*). Dashed curves corresponding to the competitive binding data for wildtype enzyme (*Figure 1B*) are shown for reference. Numerical data are reported in *Figure 5—source data 3*.

DOI: https://doi.org/10.7554/eLife.31724.022

The following source data and figure supplement are available for figure 5:

**Source data 1.** Binding affinities of Top6B mutants for different length duplexes.
DOI: https://doi.org/10.7554/eLife.31724.024
**Source data 2.** Affinities of H2TH and KGRR mutants for supercoiled and sheared salmon-sperm DNA as compared to wild type presented in *Figure 1*.
DOI: https://doi.org/10.7554/eLife.31724.025
**Source data 3.** Numerical data associated with *Figure 5*.

*Figure 5 continued on next page*

*Figure 5 continued*

DOI: https://doi.org/10.7554/eLife.31724.026

**Figure supplement 1.** Affinities of H2TH^AAA and H2TH^EEE topo VI mutants for short, defined duplexes as determined by competitive binding.

DOI: https://doi.org/10.7554/eLife.31724.023

wild type, with the H2TH^AAA mutant taking on a more open state, and the KGRR^AAA mutant taking on a more closed state. Although the H2TH^AAA mutant displayed a lower FRET signal than wild type, both in the presence of supercoiled DNA alone and with supercoiled DNA and nucleotide, the addition of AMPPNP produced a rapid FRET increase in the H2TH^AAA mutant similar to that of native topo VI, indicating that the ATPase region of this mutant responds to supercoiled DNA and nucleotide in a wildtype-like manner. By contrast, the KGRR^AAA mutant initially manifested a higher FRET signal than either wildtype topo VI or the H2TH^AAA mutant in the presence of supercoiled DNA alone; however, the addition of AMPPNP failed to elicit any further increase in FRET (*Figure 6D*). Given that ATP binding and hydrolysis rely on Top6B dimerization, and that the maximum observed ATPase rate of the KGRR^AAA construct is actually greater than wild-type topo VI in the presence of supercoiled DNA (*Figure 4B*), the high initial FRET signal for this mutant suggests that its Top6B subunits can adopt a 'pre-dimerized' ATPase competent state in the presence of supercoiled DNA alone. As a consequence, the rapid ATP turnover by the KGRR mutants likely arises from the decoupling of ATP hydrolysis and product release from a slow conformational change necessary for strand passage (i.e. those that drive G-segment opening and T-segment release). In this view, the KGRR loop would serve not only as a sensor of DNA crossings, but also as an element that delays ATP turnover until T-segment binding or strand passage has occurred.

## The H2TH interface engages an extended G-segment to couple nucleotide-dependent Top6B dimerization with DNA cleavage

Since the H2TH domain does not appear to participate in T-segment sensing (*Figure 6D*), yet is important for the strand passage activity of topo VI (*Figure 4A*), we considered whether this element might instead interact with the G-segment. The H2TH domains reside far from the site of G-segment cleavage in the Top6A dimer (*Corbett et al., 2007*; *Graille et al., 2008*); however, a prior AFM study has reported that topo VI can bend DNA by 100–140° (*Thomson et al., 2015*). Modeling DNAs with varying bend angles into structures of *S. shibatae* topo VI, which was captured in a splayed-open B-subunit conformation (*Graille et al., 2008*), suggested that a ~70 bp duplex with a ~100° bend could span both H2TH domains in a topo VI holoenzyme, running along the helical Stalk/WKxY region of the Top6B transducer domains and through the Top6A catalytic center (*Figure 7A*). Based on this model, a ~30 bp duplex would fully engage the Top6A dimer and one Stalk/WKxY element, whereas a ~40 bp duplex would be sufficient to span both Stalk/WKxY elements in a Top6A/Top6B heterotetramer (*Figure 7B*). An extended G-segment interface of this nature would not only provide a physical rationale for the marked increase in affinity of topo VI for duplex DNA as substrate length is increased from 20 to 30 bp (*Figure 1A*), but also would account for the observed DNA binding deficiencies exhibited by the Stalk/WKxY mutants (*Figure 5A*). Similarly, the impaired DNA binding of the KGRR^AAA mutant (*Figure 5—source data 1*) may reflect its apparent altered conformational state (*Figure 6—figure supplement 1*), which might misalign the G-segment-binding surfaces of the B and A subunits to lower the affinity of the enzyme for duplexes that are not already pre-bent.

If the H2TH domains do engage G-segment DNAs, they do not contribute appreciably to DNA binding, at least as judged by the affinity of the 70 bp duplex for wildtype topo VI compared to shorter duplexes (*Figure 1A*). We therefore considered whether the H2TH domains might instead help bend DNA, serve as sensors for pre-bent substrates, and/or help couple B-subunit dimerization to G-segment cleavage or strand passage. To test these ideas, we first assessed the minimal length of DNA required for nucleotide-dependent G-segment cleavage. Topo VI was incubated with 40, 60, or 70 bp long 5'-labeled duplexes in the absence of nucleotide, or with ATP or AMPPNP. Reactions were analyzed by denaturing urea-formamide PAGE to separate cleaved and uncleaved oligonucleotide products. Although the absence of a T-segment strongly inhibits G-segment scission, topo VI produced clear cleavage products in the presence of either ATP or AMPPNP on the 70 bp

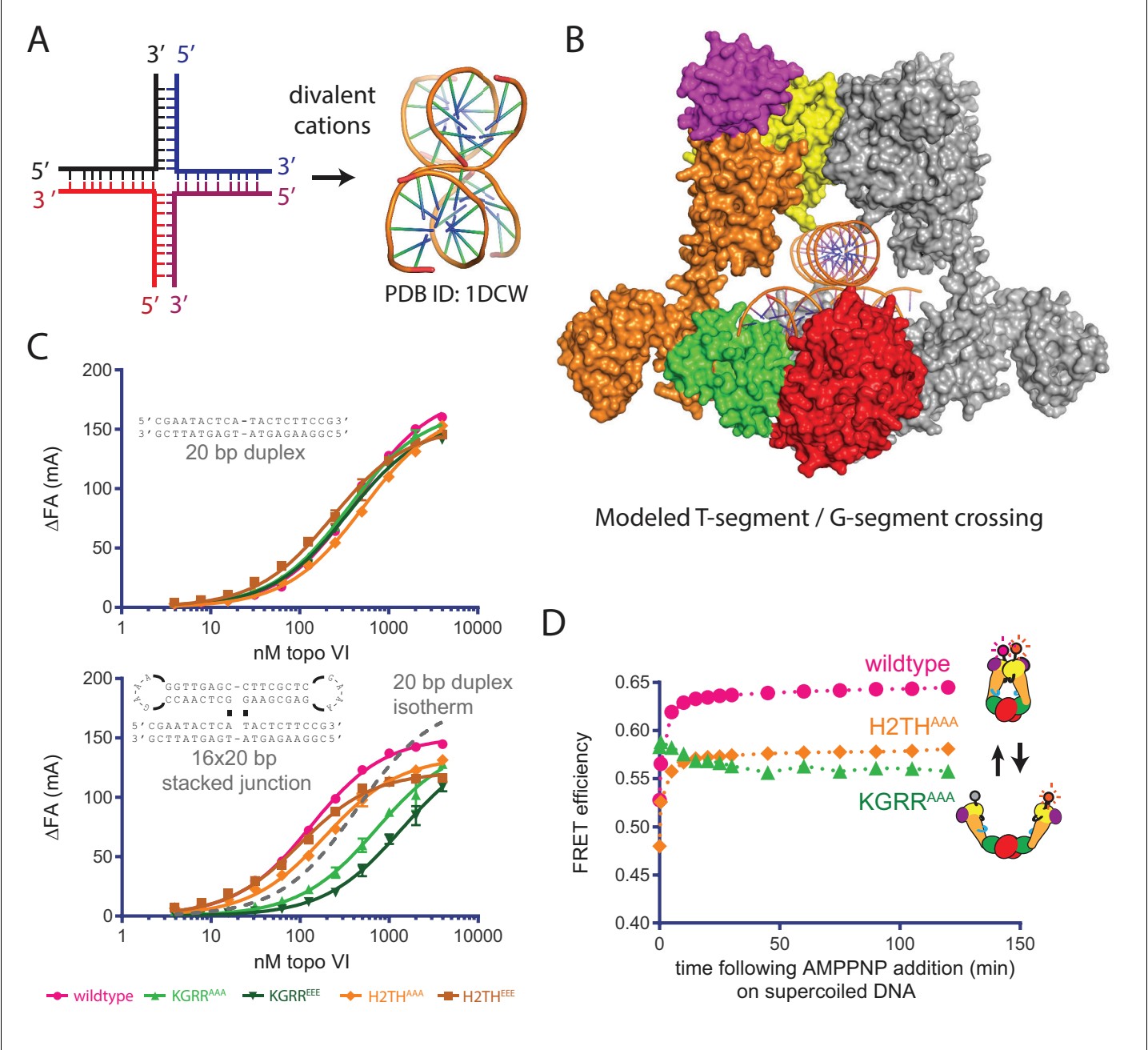

**Figure 6.** Effect of neutralization and charge reversal mutations to the KGRR loop or H2TH domain on DNA crossing affinity and Top6B dimerization. (**A**) A four-way junction folds into a stacked-X structure in the presence of divalent cations (PDB ID 1DCW) (*Eichman et al., 2000*). (**B**) Modeling of a prospective G-segment and T-segment DNA into a previously published structure of *M. mazei* topo VI (PDB ID 2Q2E)(*Corbett et al., 2007*). Domains are colored as in *Figure 3A*. The juxtaposition of the two DNAs in this intermediate closely mimic the stacked-X junction structure in *Figure 6A*. (**C**) Binding of a 20 bp fluorescein-labeled duplex (*top*) or 20 bp by 16 bp fluorescein-labeled stacked junction substrate (*bottom*, both 20 nM) to topo VI or mutant constructs. Binding was observed as a change in fluorescence anisotropy (ΔFA) and measured in milli-anisotropy units (mA) as a function of enzyme concentration. Points and bars correspond to the mean and error of three independent experiments. Curves represent fits to a single site ligand depletion binding model. In the plot of the enzyme-stacked junction binding isotherms, the fit of wildtype topo VI binding to the 20 bp duplex is displayed (—) for reference. Apparent dissociation constants are reported in *Figure 6—source data 1*. (**D**) Change in ratiometric FRET efficiency for the indicated Alexa555/647-labeled topo VI constructs incubated with supercoiled DNA was monitored over time following the addition of AMPPNP. As further detailed in *Figure 6—figure supplement 1*, incubation with supercoiled DNA alone increases the FRET efficiency for each construct. Numerical data are reported in *Figure 6—source data 2*.

DOI: https://doi.org/10.7554/eLife.31724.027

*Figure 6 continued on next page*

*Figure 6 continued*

The following source data and figure supplement are available for figure 6:

**Source data 1.** Affinities of wildtype, H2TH and KGRR mutants for stacked junction DNA.
DOI: https://doi.org/10.7554/eLife.31724.029
**Source data 2.** Numerical data associated with *Figure 6*.
DOI: https://doi.org/10.7554/eLife.31724.030
**Figure supplement 1.** Conformational response of H2TH[AAA] and KGRR[AAA] constructs on different substrates as determined by FRET in the absence of nucleotide.
DOI: https://doi.org/10.7554/eLife.31724.028

duplex. Faint cleavage products were also produced from the 60 bp duplex, but only in the presence of AMPPNP. The length of the cleavage products suggest *M. mazei* topo VI is cutting DNA slightly off-center from the preferred site identified for its *S. shibatae* homolog; these products are instead consistent with strand scission occurring at a secondary site six nucleotides upstream of this locus (*Buhler et al., 2001*). No cleavage was seen for any condition on the 40 bp duplex (*Figure 7C*).

We next assessed whether the H2TH domains play a role in the observed length dependence of the G-segment cleavage reaction by measuring the nucleotide-dependent cleavage activity of our functional mutant panel on a 70 bp duplex (*Figure 7C*). The KGRR mutants showed a slight decrease in AMPPNP-dependent cleavage, while the two Stalk/WKxY mutants displayed a greater decrease in this activity (the triple glutamate substitution proved the most severely compromised). These results are consistent with the impaired affinities that these mutant enzymes show for the 70 bp substrate (*Figure 5A*). By contrast, neither H2TH mutant proved capable of supporting short duplex cleavage. Collectively, these findings support the idea that for a G-segment to bind productively to the Top6A dimer, it ideally should be sufficiently long to engage both the stalk and H2TH regions of Top6B. The inability of a 40 bp duplex to support cleavage, even though this DNA binds with higher affinity than a 20 bp duplex and is long enough to reach both Stalk/WKxY regions, suggests G-segment DNAs must engage at least one H2TH domain before strand scission can be triggered.

One implication of H2TH contacts with the distal arms of an associated G-segment is that ATP-binding and ATPase domain dimerization might in turn alter G-segment bending. To test this prediction, we labeled opposing ends of the 70 bp duplex with Cy5 and Cy5.5 and monitored changes in the end-to-end distance by FRET for native topo VI and our panel of mutants. Bulk FRET efficiencies in the absence and presence of enzyme were measured by exciting Cy5 at 630 nm and scanning the spectral emission of both the donor and acceptor fluorophores (*Figure 7D*). The time-dependent conformational response to the addition of AMPPNP was also assessed. The addition of wild-type topo VI alone to the labeled DNA led to a modest FRET increase, a result indicative of G-segment bending that accords with prior AFM data (*Thomson et al., 2015*). The KGRR mutants produced a similar FRET increase; however, both sets of Stalk/WKxY and H2TH mutants yielded only a minor nucleotide-independent response (between that of wildtype topo VI and the free duplex). Upon adding AMPPNP, FRET efficiency rapidly increased further for the labeled DNA incubated with topo VI, or the KGRR[AAA] or KGRR[EEE] mutants, indicating that nucleotide-driven dimerization of the ATPase regions leads to additional DNA bending. This FRET increase did not occur when Cy5 and Cy5.5 were placed on separate duplexes (*Figure 7—figure supplement 1*), allowing us to attribute the observed changes in FRET with the doubly labeled DNA to intramolecular bending, rather than the binding of two segments *in trans*. While this result for the KGRR constructs initially appeared to contradict the inability of nucleotide to alter Top6B conformation in the KGRR[AAA] mutant on supercoiled DNA (*Figure 6D*), we note that the substrate differed between these two experiments. Performing the Top6B dimerization experiment with excess, unlabeled 70 bp duplex showed that, similar to wildtype topo VI bound to linear DNA, the KGRR[AAA] mutant adopts a more open conformation when bound to the 70 bp duplex than when bound to supercoiled DNA, and that the addition of nucleotide can shift the conformational equilibrium of the enzyme toward a closed state (*Figure 7—figure supplement 2*). For their part, both Stalk/WKxY mutants produced a FRET increase in the presence of AMPPNP, albeit with substantially slowed kinetics that likely account for their negligible ATPase activities (*Figure 4B*). By contrast, the H2TH mutants did not support any nucleotide-dependent increase in FRET, indicating that Top6B dimerization in these mutants no longer introduces DNA bending to the distal ends of a bound G-segment. Together, these observations

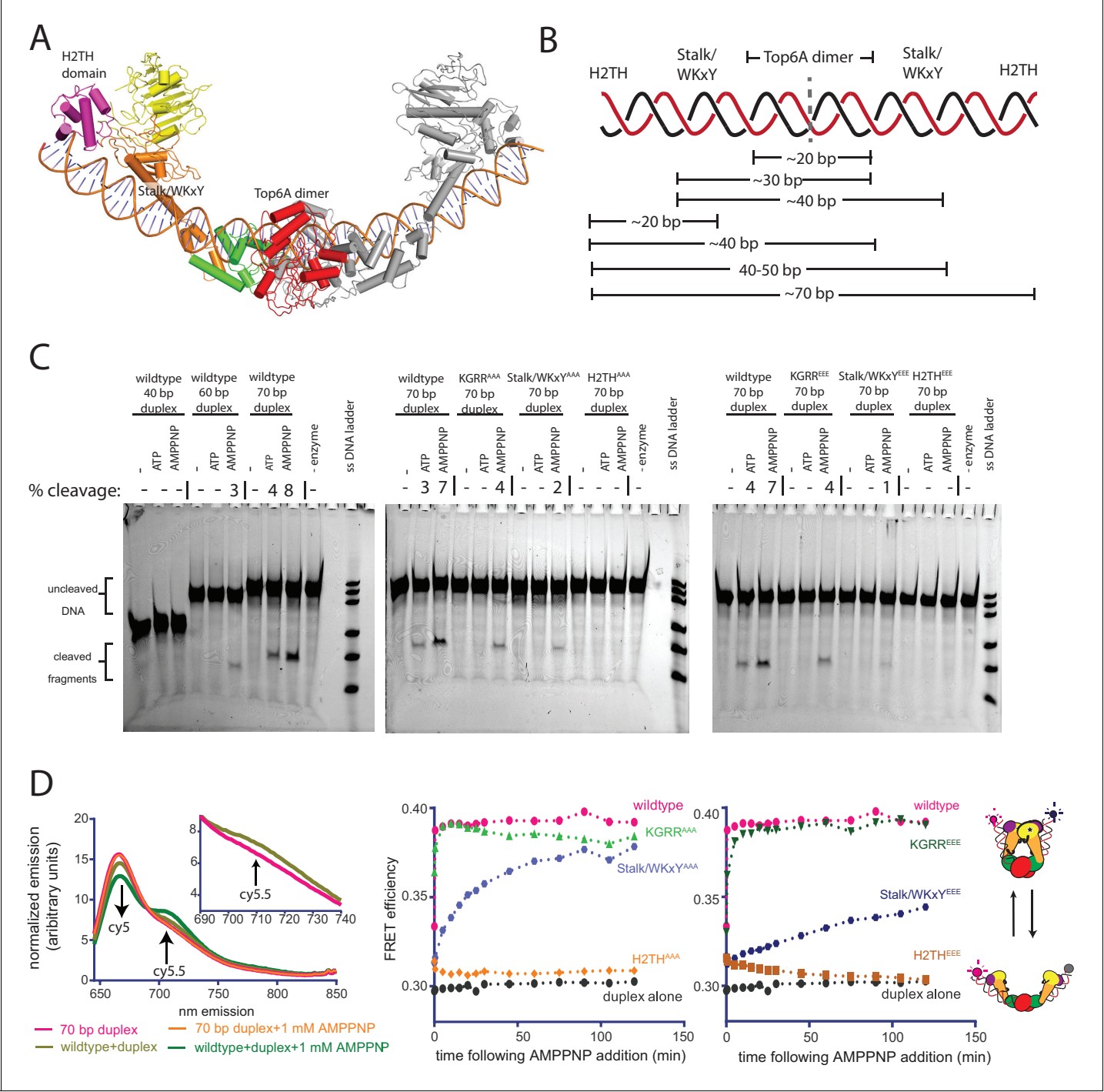

**Figure 7.** Topo VI requires H2TH-mediated, nucleotide-dependent bending of a 70 bp duplex G-segment to induce cleavage. (**A**) Model of a 70 bp bent duplex which spans dimer-related H2TH domains through the TOPRIM/Winged Helix Domain cleavage site of Top6A (using a previously published SAXS model of *S. shibatae* topo VI with Top6B in an open conformation (*Corbett et al., 2007*) see also PDB ID: 2ZBK for a similar conformation, stabilized by the inhibitor radicicol (*Graille et al., 2008*)). Domains are colored as in *Figure 3A*. DNA was modeled with a continuous bend using web 3DNA (*Zheng et al., 2009*). (**B**) Schematic of the estimated duplex lengths needed to span across the H2TH, Stalk/WKxY, and Top6A dimer DNA-binding regions, using the G-segment path modeled in *Figure 7A*. Note that the Stalk/WKxY region may allow for the asymmetric binding of DNA in different registers, accounting for the jump in affinity seen between 20 and 30 bp DNA duplexes. (**C**) Nucleotide-dependent cleavage of fluorescein-labeled DNA duplexes by topo VI and mutant constructs. Length-dependent cleavage by wildtype (*left*), cleavage of a 70 bp duplex by basic-to-neutral mutants (*middle*), and cleavage by basic-to-acidic mutants (*right*) was tested. Cleavage reactions containing a 2:1 ratio of enzyme: duplex were run on denaturing PAGE to separate reaction products, and were visualized using a laser gel scanner. Enzyme construct (wildtype,

*Figure 7 continued on next page*

*Figure 7 continued*

KGRR^AAA, KGRR^EEE, Stalk/WKxY^AAA, Stalk/WKxY^EEE, H2TH^AAA, or H2TH^EEE), duplex length (40 bp, 60 bp, or 70 bp), and addition of 1 mM ATP or 1 mM AMPPNP is noted above each lane. A no enzyme control containing 1 mM AMPPNP and a single strand DNA ladder consisting of 20, 30, 40, 60, 70, and 80 nt oligonucleotides were run for reference. Where present, the percentage of cleavage product relative to intact DNA is quantified above the lane. (D) Nucleotide-dependent bending of a Cy5/Cy5.5-labeled 70 bp duplex was assessed using bulk FRET. Fluorescence emission spectra (*left*) produced by 630 nm excitation of the Cy5-Cy5.5-labeled DNA show an increase in cy5.5 emission in the presence of topo VI (*left, inset*) and AMPPNP, but not in the presence of AMPPNP alone. Spectral emission was normalized by total emission from 645 nm to 850 nm. Ratiometric FRET efficiency was monitored over time upon addition of AMPPNP for the noted basic-to-neutral mutant (*middle*) or basic-to-acidic topo VI mutant (*right*). Wildtype and duplex alone are shown in each case for comparison. *Figure 7—figure supplement 1* confirms FRET changes arise from DNA bending. *Figure 7—figure supplement 2* further considers the gate closure activity of KGRR^AAA on the 70 bp duplex substrate. Numerical data are reported in *Figure 7—source data 1*.

DOI: https://doi.org/10.7554/eLife.31724.031

The following source data and figure supplements are available for figure 7:

**Source data 1.** Numerical data associated with *Figure 7*.
DOI: https://doi.org/10.7554/eLife.31724.034
**Figure supplement 1.** The AMPPNP-dependent FRET increase observed for a labeled 70 bp duplex arises from DNA bending by topo VI.
DOI: https://doi.org/10.7554/eLife.31724.032
**Figure supplement 2.** AMPPNP-dependent conformational response of KGRR^AAA in bending assay conditions bound to excess 70 bp duplex or supercoiled DNA.
DOI: https://doi.org/10.7554/eLife.31724.033

both suggest that topo VI engages a G-segment using an extended interface that runs from one H2TH domain to the other, and that strand scission is stimulated by bending induced by Top6B dimerization. The observation that the ATPase activity of the H2TH mutants is decoupled from strand passage, yet also substantially impaired when compared to the futile cycling of the KGRR mutants (*Figure 4*), further suggests that there is a feedback mechanism which couples nucleotide turnover to efficient G-segment deformation and cleavage.

## Discussion

### The ATPase elements of type IIB topoisomerases engage supercoiled DNA to regulate DNA strand passage

Using a broad range of functional and reporter assays (summarized in *Tables 1–2*), we show here that type IIB topoisomerases preferentially engage the DNA crossings and bends of supercoiled substrates, and that binding to supercoiled DNA in turn stimulates the nucleotide-dependent dimerization of Top6B and couples this movement to DNA cleavage and strand passage at a distance in Top6A. To recognize and exploit distinguishing features of supercoiled substrates, topo VI uses several previously unidentified DNA-binding elements integrated into Top6B, including: (1) a basic interface formed along the subunit's C-terminal stalk and a conserved WKxY motif that is important for robust G-segment binding (*Figure 3E*, *Figure 5A*), (2) a basic 'KGRR' loop in the GHKL domain that aids DNA crossing recognition and links controlled ATP turnover to productive strand passage (*Figure 3D*, *Figure 4*, *Figure 6*), and (3) an H2TH DNA-binding domain that promotes nucleotide-dependent G-segment bending and links ATP turnover to DNA cleavage and strand passage (*Figure 3F*, *Figure 7C–D*). Collectively, our data highlight new intermediate steps in the topo VI catalytic cycle (*Figure 8*) and provide a molecular rationale for the essential role of Top6B in driving transesterase activity by Top6A (*Buhler et al., 1998*; *Buhler et al., 2001*). By demonstrating that efficient and productive Top6B dimerization requires nucleotide, supercoiled DNA, and an intact KGRR loop, our findings also suggest that the previously visualized, inactive conformation of the Top6A dimer (*Corbett et al., 2007*; *Graille et al., 2008*; *Nichols et al., 1999*) may represent a cleavage-suppression mechanism that can only be overcome when the regions identified here are occupied by the binding of an extended DNA crossing and when nucleotide induces the dimerization of Top6B.

Besides promoting Top6B closure, our data also imply that T-segment engagement may actively control both ATP turnover and DNA-gate opening to permit strand passage. As with wild-type topo VI, the binding of the KGRR^AAA mutant to supercoiled DNA alone promotes Top6B dimerization

**Table 1.** Summary of topo VI functional activities and mutant effects.

| | | Effect of mutation to: | | |
|---|---|---|---|---|
| Assay | Wildtype activity | KGRR loop | Stalk/WKxY | H2TH |
| Supercoil relaxation | distributive strand passage activity | kills strand passage | kills strand passage | greatly impairs strand passage |
| ATPase activity | | | | |
| -on linear DNA | stimulates above basal activity | no activity | no activity | ~wildtype activity |
| -on supercoiled DNA | stimulates more than linear DNA | increased activity, futile cycling | no activity | ~wildtype activity, futile cycling |
| DNA binding | | | | |
| -short duplexes | affinity increases from 20 bp to 40 bp in length | moderately impairs binding for longer duplexes | greatly impairs binding | slightly impairs binding for longer duplexes |
| -sheared salmon-sperm DNA | similar affinity as for 40–70 bp duplexes | moderately impairs binding | N.D. | slightly impairs binding |
| -supercoiled DNA | increased affinity compared to linear DNA | moderately impairs binding | N.D. | moderately impairs binding |
| -stacked junction | tighter binding than to duplex | greatly impairs binding | N.D. | ~wildtype affinity |
| Top6B dimerization | | | | |
| -on short DNA duplexes | DNA promotes closure AMPPNP promotes further closure | loss of substrate promoted closure AMPPNP promotes some closure | N.D. | N.D. |
| -on supercoiled DNA | promotes greater closure than linear DNA AMPPNP promotes further closure | supercoiled DNA promotes closure loss of AMPPNP promoted closure | N.D. | weaker substrate dependent closure than wildtype AMPPNP promotes further closure |
| Short duplex cleavage | AMPPNP promotes cleavage on 60 and 70 bp duplexes | similar to wildtype | greatly impairs cleavage | no cleavage |
| Short duplex bending | AMPPNP promotes bending | similar to wildtype | greatly slows bending | no bending |

DOI: https://doi.org/10.7554/eLife.31724.037

**Table 2.** Summary of enzyme, DNA and nucleotide conditions by type of experiment.

| Assay | [enzyme] | [nucleotide] | [DNA] |
|---|---|---|---|
| DNA binding | 0, 3.9–4000 nM | N/A | 20 nM (0.4–1.4 µM bp) probe duplex |
| Competitive binding | 100 nM | N/A | 20 nM (1.4 µM bp) probe duplex 0, 0.1–106 µM bp DNA competitor/ 0, 0.3–36 nM plasmid |
| Supercoil relaxation *titration* *timecourse/chase* | 0, 0.3-20 nM 2.5 nM | 1 mM ATP 1 mM ATP | *2.9 kb primary plasmid-* 10.2 µM bp DNA/3.5 nM plasmid *6.5 kb chase plasmid-* 10.2 µM bp DNA/1.6 nM plasmid |
| ATP hydrolysis- *ATP titration* *DNA titration* | 500 nM 500 nM | 0, 0.06–4 mM ATP 2 mM ATP | 400 µM bp DNA/136 nM plasmid 0, 3.1–800 µM bp DNA/ 0, 1–273 nM plasmid |
| Top6B dimerization | 200 nM | 1 mM AMPPNP | 100 µM bp DNA/34 nM plasmid |
| Short duplex cleavage | 200 nM | 1 mM ATP or AMPPNP | 100 nM duplex (7 µM bp DNA) |
| Short duplex bending | 200 nM | 1 mM AMPPNP | 100 nM duplex (7 µM bp DNA) |

DOI: https://doi.org/10.7554/eLife.31724.038

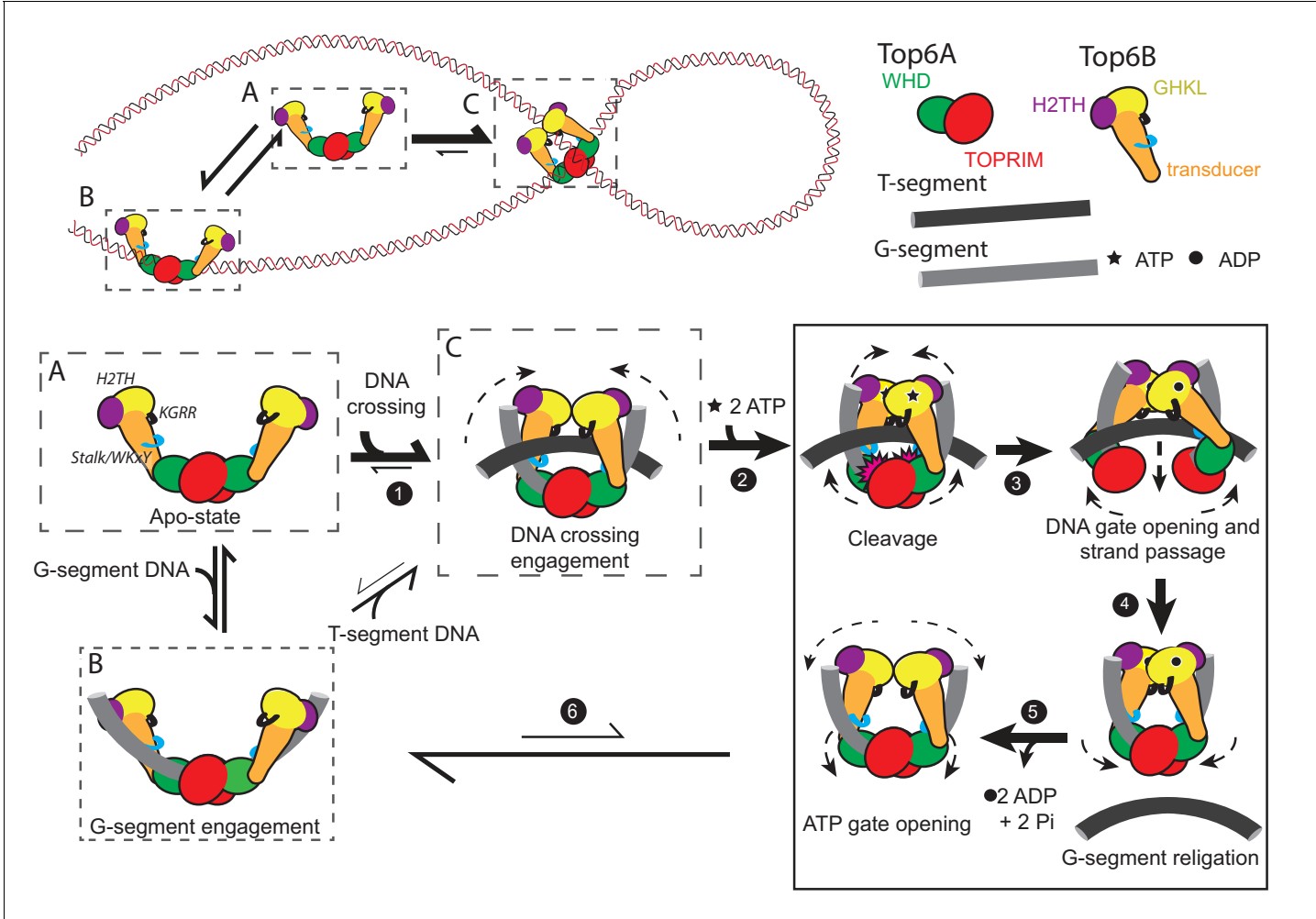

**Figure 8.** A new model for the Type IIB topoisomerase catalytic cycle. Free topo VI (A) binds to linear DNA (B), but preferentially engages DNA crossings (C). Binding to a hooked DNA crossing (C) by the KGRR loop and Stalk/WKxY region (*step 1*) induces a conformational change that presets Top6B dimerization. From this state ATP binding (*step 2*) introduces H2TH-dependent G-segment DNA bending and shifts the catalytic tyrosines on the Winged Helix Domain (WHD) into a cleavage-competent conformation, thus committing the enzyme to strand passage and ATP hydrolysis. While the KGRR/T-segment interaction stabilizes Top6B dimerization, T-segment capture potentiates DNA-gate opening by introducing strain in the storage cavity (*step 3*). T-segment release allows for DNA-gate closure and G-segment religation (*step 4*). Without a DNA crossing to stabilize Top6B closure, ADP and $P_i$ are released, the WHDs relax to an inactive conformation (*step 5*) and Top6B returns to a relaxed, open conformation (*step 6*). From this G-segment bound state (B) topo VI tends to dissociate from DNA (to state A), but will infrequently capture another T-segment, regenerating a DNA crossing (C). The mechanistic implications of this model for meiotic recombination systems are considered in *Figure 8—figure supplement 1*.

DOI: https://doi.org/10.7554/eLife.31724.035

The following figure supplement is available for figure 8:

**Figure supplement 1.** Conservation of canonical GHKL elements between Top6B, MTop6B, and Top6BL.

DOI: https://doi.org/10.7554/eLife.31724.036

(*Figure 6—figure supplement 1*) and supports ATP hydrolysis (*Figure 4B*); however, disruption of this region impairs the binding of a DNA crossover (*Figure 6C*), does not support additional conformational response to nucleotide (*Figure 6D*), blocks ATP hydrolysis on short linear DNA (*Figure 4B*), and abolishes strand passage overall (*Figure 4A*). This behavior suggests the bends or pre-formed crossings present in supercoiled DNA help to promote B-subunit dimerization and ATP hydrolysis, and may partially compensate for a loss of the KGRR element (*Figure 4B*, *Figure 6D*), but that the coupling of ATP hydrolysis to strand passage requires the productive binding of a T-segment DNA (*Figure 2B–D*, *Figure 6D*). Given that the T-segment storage cavity appears to be too small to accommodate DNA when fully closed (*Corbett et al., 2007*; *Corbett and Berger, 2005*;

*Graille et al., 2008*), it has been proposed that T-segment engagement may potentiate opening of the Top6A dimer and separation of a cleaved G-segment. Considering this idea in light of our present findings suggests that the binding of the KGRR loops to a stored T-segment helps to suppress premature release of ATP hydrolysis products, which is normally linked to a slow conformational change associated with G-segment separation and subsequent T-segment release. This scheme offers a simple explanation for why the KGRR mutants rapidly hydrolyze ATP when bound to super-coiled DNA: weakening of the T-segment interaction with the KGRR loop allows for early ATP turn-over, yet by not resolving the DNA crossing, Top6B remains pre-dimerized and primed to bind ATP again, leading to futile cycling without strand passage.

## Topo VI's enzymatic properties seem oddly mismatched to the expected demands of the cell

Certain biochemical properties of topo VI identified here are somewhat surprising when considering the demands placed on the cell by transcription and replication. For example, the highly distributive nature of supercoil relaxation observed for topo VI (*Figure 1C–D*) is at odds with a need to remove the continual local build-up of superhelical tension arising from RNA polymerase advancement or replication fork progression. The maximal observed ATP hydrolysis and strand passage rates for *Mm* topo VI in vitro (*Figure 1C*, *Figure 2A*) are also much slower (~50–100 fold) than rates generally observed for type IIA topoisomerases (*Higgins et al., 1978*; *Lindsley and Wang, 1993*; *Osheroff et al., 1983*; *Sugino and Cozzarelli, 1980*). These enzymatic properties raise important questions as to when and in which context topo VI acts in the cell. For instance, using estimates based on the genome size and generation time of *M. mazei* (Appendix 1), topo VI would appear to require ~50-fold greater specific activity to keep up with gene expression and chromosome duplication, or else be present at extremely high cellular concentrations.

Although the source of this discrepancy may be due to differences between in vitro vs. in vivo rates of strand passage, it could alternatively arise from a missing factor that enhances topo VI activity. This second explanation, if true, has intriguing ramifications. For example, if a secondary factor were to accelerate topo VI's strand passage rate by increasing processivity, then the distributive action of topo VI might reflect an auto-inhibitory mechanism that is manifest until the enzyme is localized to the appropriate chromosomal context. Along this line, multiple protein factors have been identified to bind topo VI in *Arabidopsis thaliana*, and may be obligate components of the topo VI machinery in plants (*Breuer et al., 2007*; *Forterre and Gadelle, 2009*; *Kirik et al., 2007*; *Sugimoto-Shirasu et al., 2005*). If analogous factors exist for archaeal topo VI, it may be that Top6A and Top6B actually constitute the core of a larger type IIB topoisomerase complex. Eukaryotic topo IIIα, a type IA topoisomerase, exemplifies such a strategy, interacting with a RecQ family helicase and SSB/RPA-like factors to channel DNA strand passage into efficient Holliday junction resolution (*Plank et al., 2006*; *Raynard et al., 2006*; *Singh et al., 2008*; *Wu et al., 2006*; *Xu et al., 2008*).

## Implications of functional features of type IIB topoisomerases for homologous ATPase, nuclease, and transesterase systems

The picture of the type IIB topoisomerase strand passage mechanism developed here reveals a rich set of regulatory mechanisms both shared with and divergent from type IIA topoisomerases. Of these, a central feature is the requirement that topo VI must engage a DNA crossing (as found in supercoiled or catenated DNA) to access a stable dimerized B-subunit conformation and productively turn over ATP (*Figure 8*). Contrariwise, Top6B takes on a predominantly open conformation when bound only to a prospective G-segment (or in the absence of DNA [*Figure 2*, *Corbett et al., 2007*; *Graille et al., 2008*]), even when nucleotide is present. This tight control over Top6B dimerization, and its reciprocal coupling to G-segment cleavage, likely helps compensate for the missing safeguard of a third dimerization interface – the so-called 'C-gate' – found in type IIA topoisomerases (*Roca et al., 1996*; *Roca and Wang, 1994*; *Williams and Maxwell, 1999*). For its part, the ATPase region of type IIA topoisomerases does possess potential T-segment-sensing elements (*Tingey and Maxwell, 1996*); however, in contrast to topo VI, the ATPase domains of type IIA topoisomerase holoenzymes readily dimerize upon binding ATP, even when DNA is absent (*Gubaev and Klostermeier, 2011*; *Roca and Wang, 1992*). Interestingly, in requiring the binding of a substrate

T-segment for stable ATPase domain dimerization, topo VI echoes the behavior of Hsp90, whose related GHKL ATPase fold strongly depends on client protein or co-chaperone engagement to drive ATPase association (*Ali et al., 2006*; *Hessling et al., 2009*; *Wolmarans et al., 2016*). This co-dependency raises the possibility that other GHKL ATPases, such as MutL and MORC proteins, may similarly rely on substrate/cofactor interactions as a checkpoint to license nucleotide-dependent dimerization.

Following DNA crossover recognition, ATP binding by Top6B is needed to trigger G-segment scission by Top6A (*Buhler et al., 1998*). However, closing of the ATP gate also serves to further bend the G-segment through contacts mediated by the H2TH domain (*Figure 7D*). Although H2TH mutants are unable to cleave short duplex substrates (*Figure 7C*), they support weak strand passage activity on supercoiled DNA (*Figure 4A*), a substrate that constrains DNA bends independent of enzyme binding. This suggests that DNA bending itself, whether innate or H2TH-mediated, may help promote DNA breakage by Top6A. Although the H2TH domain is specific to type IIB topoisomerases, type IIA enzymes also bend G-segment DNAs to support cleavage (*Dong and Berger, 2007*; *Laponogov et al., 2009*; *Lee et al., 2013*; *Lee et al., 2012*; *Wohlkonig et al., 2010*). This dependency raises the possibility that other nucleases or transesterases that rely on the TOPRIM fold beside type II topoisomerases (e.g. OLD family enzymes and Spo11 (*Aravind et al., 1998*)) may similarly require DNA deformation to promote strand scission.

The discovery that Spo11 was related to the DNA-cleaving Top6A subunit of archaeal topo VI was a critical development in understanding how DNA breaks are formed to initiate meiotic recombination (*Bergerat et al., 1997*; *Keeney et al., 1997*). The realization that Top6A requires Top6B for DNA cleavage (*Buhler et al., 1998*) has in turn raised the question of whether Spo11 might partner with a similar regulatory factor during meiotic recombination. Recently, structurally homologous counterparts to Top6B have been recognized across a wide range of eukaryotic species (MTop6B in plants, Top6BL in mammals, Rec102 in *S. cerevisiae*, and Mei-P22 in *Drosophila*) (*Robert et al., 2016*; *Vrielynck et al., 2016*). Interestingly, the WKxY motif implicated here in G-segment binding is conserved between Top6B and some of its meiotic homologs (e.g. mammalian MTop6B and plant Top6BL) (*Robert et al., 2016*), suggesting that this region could assist Spo11 with DNA targeting, and contribute to the signals necessary to activate DNA cleavage during meiosis. In those Top6B homologs where the WKxY motif is poorly conserved, alternative features on the transducer stalk may participate in binding to DNA. For example, the prospective WKxY motif in budding yeast Rec102 is highly divergent in sequence (WEEQ), yet Spo11 hotspots from this organism display a sequence bias that extends beyond the predicted footprint of the Spo11 dimer. Interestingly, this bias maps to a distance of ±11–16 bp from the dyad of Spo11 (*Pan et al., 2011*), compared to the ~17–20 bp distance between the Top6B Stalk/WKxY region and the Top6A dyad, consistent with the notion that non-Spo11 DNA interaction sites may have shifted during evolution.

In topo VI, we find that Top6B dimerization further bends DNA to potentiate cleavage by Top6A. Surprisingly, components critical for Top6B-mediated dimerization are either highly divergent or missing in meiotic Top6B homologs. For example, both Topo6BL and MTopo6B contain a highly degenerate GHKL domain that lacks essential elements required for ATP binding (only purine-binding elements are conserved, see *Figure 8—figure supplement 1*), and Rec102 and Mei-P22 lack a GHKL domain entirely (*Dutta and Inouye, 2000*; *Robert et al., 2016*; *Vrielynck et al., 2016*). Insofar as DNA bending, the meiotic Top6B-like factors identified thus far also lack an H2TH domain (*Robert et al., 2016*; *Vrielynck et al., 2016*). Should Spo11, like Top6A, require both DNA bending and allosteric activation to achieve a cleavage-competent state, these differences indicate that it is not the newly identified Top6B-like subunits alone that are responsible for mediating this event. Candidate factors that might further regulate Spo11-dependent break formation include additional partner proteins, post-translational modifications, and tension on or deformation of the DNA itself by factors responsible for sister chromatid pairing (*Lam and Keeney, 2014*). Future studies focused on defining how topo VI and Spo11-type systems physically engage DNA strands, respond to possible partner factors, and switch between inactive and active DNA-cleavage states will be needed to help shed light on how these systems operate.

# Materials and methods

## Key resources table

| Reagent type (species) or resource | Designation | Source or reference | Identifiers | Additional information |
|---|---|---|---|---|
| Gene (*Methanosarcina Mazei*) | Top6A | N/A | NCBI Gene ID: 1480760 | |
| Gene (*Methanosarcina Mazei*) | Top6B | N/A | NCBI Gene ID: 1480759 | |
| Strain, strain background (*E. coli*) | BL21(DE3)-RIL | QB3-MacroLab | | |
| Strain, strain background (*E. coli*) | XL1-Blue | QB3-MacroLab | | |
| Recombinant DNA reagent | *M. mazei* Top6AB expression vector | PMID: 17603498 | | |
| Recombinant DNA reagent | *M. mazei* Top6AB-KGRR$^{AAA}$ expression vector | this paper | | Construct generated by introduction of point mutations: K186A, R188A, and R189A to Top6B gene on *M. Mazei* Top6AB expression vector |
| Recombinant DNA reagent | *M. mazei* Top6AB-KGRR$^{EEE}$ expression vector | this paper | | Construct generated by introduction of point mutations: K186E, R188E, and R189E to Top6B gene on *M. Mazei* Top6AB expression vector |
| Recombinant DNA reagent | *M. mazei* Top6AB-Stalk/WKxY$^{AAA}$ expression vector | this paper | | Construct generated by introduction of point mutations: K399A, K401A, and R457A to Top6B gene on *M. Mazei* Top6AB expression vector |
| Recombinant DNA reagent | *M. mazei* Top6AB-Stalk/WKxY$^{EEE}$ expression vector | this paper | | Construct generated by introduction of point mutations: K399E, K401E, and R457E to Top6B gene on *M. Mazei* Top6AB expression vector |
| Recombinant DNA reagent | *M. mazei* Top6AB-H2TH$^{AAA}$ expression vector | this paper | | Construct generated by introduction of point mutations: R263A, K268A, and K308A to Top6B gene on *M. Mazei* Top6AB expression vector |
| Recombinant DNA reagent | *M. mazei* Top6AB-H2TH$^{EEE}$ expression vector | this paper | | Construct generated by introduction of point mutations: R263E, K268E, and K308E to Top6B gene on *M. Mazei* Top6AB expression vector |
| Recombinant DNA reagent | *M. mazei* Top6AB-cyslite-155C expression vector | this paper | | Construct generated by introduction of point mutations: T155C, C267S, C278A, C316A, and C550A to Top6B gene on *M. Mazei* Top6AB expression vector |
| Recombinant DNA reagent | *M. mazei* Top6AB-KGRR$^{AAA}$ cyslite-155C expression vector | this paper | | Construct generated by introduction of point mutations: K186A, R188A, and R189A to Top6B gene on *M. Mazei* Top6AB-cyslite-155C expression vector |
| Recombinant DNA reagent | *M. mazei* Top6AB-H2TH$^{AAA}$ cyslite-155C expression vector | this paper | | Construct generated by introduction of point mutations: R263A, K268A, and K308A to Top6B gene on *M. Mazei* Top6AB-cyslite-155C expression vector |
| Recombinant DNA reagent | *M. mazei* Top6AB-E44A expression vector | this paper | | Construct generated by introduction of point mutations: E44A to Top6B gene on *M. Mazei* Top6AB expression vector |
| Recombinant DNA reagent | pSG483 (plasmid DNA) | PMID: 16023670 | | 2.9 kb plasmid used as supercoiled substrate |
| Sequence-based reagent (13 oligonucleotides) | See *Figure 1—source data 1* | Integrated DNA Technologies | | |
| Chemical compound, drug | salmon sperm DNA, sheared | Thermo Fisher Scientfic | ThermoFisher:AM9680 | |

*Continued on next page*

*Continued*

| Reagent type (species) or reagent | Designation | Source or reference | Identifiers | Additional information |
| --- | --- | --- | --- | --- |
| Chemical compound, drug | Alexa Fluor 555 C2 Maleimide | Thermo Fisher Scientfic | ThermoFisher:A20346 | |
| Chemical compound, drug | Alexa Fluor 647 C2 Maleimide | Thermo Fisher Scientfic | ThermoFisher:A20347 | |
| Software, algorithm | ConSurf Server | PMID: 20478830 | RRID:SCR_002320 | |
| Software, algorithm | w3DNA server | PMID: 19474339 | | |
| Software, algorithm | PyMol | Schrödinger, LLC | RRID:SCR_000305 | |
| Software, algorithm | Prism 7 | Graphpad Software | RRID:SCR_015807 | |

## Cloning of *M. mazei* topo VI functional mutant vectors

Cloning of the *M. mazei* Top6B gene in frame with an N-terminally fused His6-tobacco etch virus (TEV) protease-cleavable tag and the *M. mazei* Top6A gene into a polycistronic expression vector was previously described (*Corbett et al., 2007*). Oligonucleotides used for site directed mutagenesis were obtained from Integrated DNA Technology (IDT, Coralville, IA). Mutant constructs were generated either by PCR amplification of the expression vector using primers containing the desired point substitutions followed by blunt-end ligation, or by quick-change mutagenesis (Agilent, Santa Clara, CA). The following mutations were added to generate the 'Cys-lite' construct: C267S, C278A, C316A and C550A, all in Top6B. Mutagenesis was verified by Sanger sequencing (Genewiz LLC, South Plainfield, NJ).

## Protein expression and purification

Topo VI and functional mutant variants were overexpressed in *E. coli* BL21(DE3)Codon +RIL cells (QB3-Macrolab, University of California-Berkeley, CA) grown in ZYM-5052 auto-induction media (*Studier, 2005*). Wild-type topo VI was expressed in cultures grown at 37°C, whereas cultures expressing functional mutant constructs were shifted to 25°C upon reaching an $OD_{600}$ of 0.4–0.6. The KGRR$^{AAA}$ FRET assay construct was grown at 37°C to an $OD_{600}$ of 2–3 in M9ZB media (*Studier, 2005*), cooled to 18°C, and then induced with IPTG (250 µM final concentration) and grown overnight. Cultures were harvested by centrifugation at 24 hr following inoculation, resuspended in buffer A [20 mM HEPES-KOH pH 7.5, 800 mM NaCl, 20 mM Imidazole, 10% (v/v) glycerol, 1 µg/mL pepstatin A, 1 µg/mL leupeptin, 1 mM PMSF], and frozen drop-wise into liquid nitrogen for storage at −80°C.

Proteins were purified as previously described (*Corbett et al., 2007*). Harvested cells were lysed by sonication, and lysate was clarified by centrifugation. Clarified lysate was applied to a 5 mL HiTrap $Ni^{2+}$ column (GE Healthcare Life Sciences, Marlborough, MA, USA) and washed with buffer A [20 mM HEPES-KOH pH 7.5, 800 mM NaCl, 20 mM Imidazole, 10% (v/v) glycerol, 1 µg/mL pepstatin A, 1 µg/mL leupeptin, 1 mM PMSF]. Following a subsequent wash with buffer B [20 mM HEPES-KOH pH 7.5, 150 mM NaCl, 20 mM Imidazole, 10% (v/v) glycerol, 1 µg/mL pepstatin A, 1 µg/mL leupeptin, 1 mM PMSF], bound proteins were eluted by a 15-column volume gradient from buffer B to buffer C [20 mM HEPES-KOH pH 7.5, 150 mM NaCl, 20 mM Imidazole, 10% (v/v) glycerol, 1 µg/mL pepstatin A, 1 µg/mL leupeptin, 1 mM PMSF]. Fractions containing the topo VI heterotetramer were applied to a 5 mL HiTrap SP cation-exchange column and 5 mL HiTrap Q anion-exchange column (GE Healthcare Life Sciences) in series and washed with buffer B. The HiTrap SP column was removed, and protein bound to the HiTrap Q column was eluted with a 10-column volume gradient from buffer B to buffer A. Peak fractions were concentrated by centrifugation (Millipore Amicon Ultra 30K MWCO) and incubated with 1.5 mg of His$_6$-TEV protease (QB3-Macrolab, University of California, Berkeley) overnight at 4°C to remove His$_6$ tags. Uncleaved proteins and His$_6$-TEV protease were removed by applying the protease cleavage reaction to a HiTrap $Ni^{2+}$ column equilibrated in buffer B. Flow-through was concentrated and applied to an Sephacryl-300 HR gel filtration column (GE Healthcare Life Sciences) equilibrated and run in sizing buffer [20 mM HEPES-KOH pH 7.5, 300 mM KCl, 10% (v/v) glycerol] and concentrated by centrifugation (Millipore Amicon Ultra 10K MWCO). Purity of peak fractions was assessed by SDS-PAGE and coomassie blue staining, and the concentration of tetramer was determined by absorbance at 280 nm using extinction coefficients determined by the ExPASY ProtParam webserver (*Gasteiger et al., 2005*). Proteins were flash frozen

in a final storage buffer [20 mM HEPES-KOH pH 7.5, 300 mM KCl, 30% (v/v) glycerol, 1 mM Tris-phosphine hydrochloride (TCEP)] and stored in aliquots at −80°C for use in subsequent biochemical and biophysical studies.

## DNA binding and competition

DNA substrates were resuspended in ddH₂O and annealed from single strand DNA oligomers of complementary sequence (*Figure 1—source data 1*) obtained from IDT. Annealing of the stacked-junction DNA substrate followed published protocols (*Duckett et al., 1988*) with a few modifications. The junction was prepared in 25 mM Tris HCl pH 7.9, 25 mM NaCl, 10 mM MgCl₂ and annealed by heating at 70°C for 2 hr, followed by cooling at 0.5°C/min to 4°C. Annealing reaction products were loaded onto a 5 mL HiTrap-Q anion exchange column (GE Healthcare Life Sciences) equilibrated in stacked junction (SJ) buffer A [25 mM NaCl, 25 mM Tris-HCl pH 7.9, 10 mM MgCl₂]. Contaminants were removed by washing with 55%/45% mix of SJ buffer A to SJ Buffer B [1 M NaCl, 25 mM Tris 7.9, 10 mM MgCl₂]. Correctly annealed substrate was eluted with 45%/55% Buffer A/ Buffer B, pooled and dialyzed back into SJ buffer A, and concentrated by centrifugation (Amicon Ultra 3K MWCO, EMD Millipore, Billerica, MA). Proper annealing for all substrates was assessed by native 15% PAGE run in 0.5x Tris-Borate-EDTA (TBE) buffer at 4°C.

DNA binding by topo VI and functional mutants was assessed using fluorescence anisotropy. Protein was serially diluted in two-fold steps in binding assay dilution buffer [250 mM potassium glutamate, 5% (v/v) glycerol, 50 mM HEPES-KOH pH 7.5 and 1 mM TCEP] and incubated with fluorescein-labeled DNA substrate in the dark and on ice for 5 min. Reactions were diluted to final binding assay conditions [27 μL, 0, 0.3–4000 nM enzyme, 20 nM labeled duplex, 50 mM potassium glutamate, 5% (v/v) glycerol, 20 mM HEPES-KOH pH 7.5, 1 mM TCEP, 10 mM MgCl₂ and 0.1 mg/ mL BSA], and incubated on ice an additional 10 min. Fluorescence anisotropy was measured at ambient temperature using a Clariostar microplate reader (BMG Labtech GmbH, Ortenberg, Germany) by exciting at 482 nm (band pass 16 nm) and measuring parallel and perpendicular emission intensity at 530 nm (band pass 40 nm), with an inline 504 nm long pass dichroic filter. Data are the average of three independent experiments, with all points normalized to the DNA alone condition and fit to the following single-site binding model:

$$FA = FA_{max}\left(\frac{[L]+[P]+K_{d,app}-\sqrt{\left([L]+[P]+K_{d,app}\right)^2-4[L][P]}}{2[L]}\right) \quad (1)$$

where $\Delta FA_{max}$ is the maximal specific change in anisotropy, [L] is DNA substrate concentration, [P] is the concentration of topo VI construct, and $K_{d,app}$ is the apparent dissociation constant for DNA substrate and enzyme. To test for cooperativity, binding isotherms were also fit to a Hill equation-type model:

$$FA = FA_{max}\left(\frac{[P]^h}{K_{d,app}^h+[P]^h}\right) \quad (2)$$

where $\Delta FA_{max}$ is the maximal specific change in anisotropy, [P] is the concentration of topo VI construct, $h$ is the apparent Hill coefficient, and $K_{d,app}$ is the apparent dissociation constant for DNA substrate and enzyme.

Competition assays were carried out similarly to binding assays, with protein diluted in binding assay dilution buffer and incubated with the 70 bp fluorescein-labeled duplex and either negatively supercoiled pSG483 plasmid DNA (pBluescript SK derivative, 2927 bp) or linear sheared salmon-sperm DNA (ThermoFisher Scientific, Waltham, MA). Reactions were diluted to final binding assay conditions, except enzyme concentration was set at 100 nM, and competitor concentration varied from 0.1 μM bp to 106.5 μM bp DNA. Anisotropy data were fit to an explicit competition model (*Wang, 1995*), which fits to the parameters: [A], total concentration of the competitor DNA substrate; [B], total concentration of the labeled DNA probe; [P], total topo VI concentration; $K_A$, dissociation constant of the competitor DNA substrate; $K_B$, dissociation constant of the labeled DNA probe; and $\Delta FA_{max}$, the maximal specific change in fluorescence anisotropy for the probe.

## Supercoiled DNA relaxation

Topo VI holoenzyme was thawed and diluted in series with relaxation assay dilution buffer [300 mM potassium glutamate, 10% (v/v) glycerol, 20 mM HEPES-KOH pH 7.5 and 1 mM TCEP] and incubated with negatively supercoiled pSG483 plasmid DNA for 5 min on ice before dilution into final relaxation assay conditions [30 μL reactions, 0, 0.3–20 nM topo VI for titration, 2.5 nM topo VI for timecourses, 50 mM potassium glutamate, 10% (v/v) glycerol 20 mM bis-tris-propane-HCl (BTP-HCl) pH 7.5, 2 mM HEPES pH 7.5, 1 mM TCEP, 10 mM MgCl$_2$, 0.1 mg/mL BSA, 3.5 nM pSG483 (10.2 μM bp DNA), and 1 mM ATP]. Reactions were initiated by addition of ATP, incubated at 30°C, and quenched by addition of SDS and EDTA to final concentrations of 1% and 10 mM respectively. Glycerol-based loading dye was added to samples which were run on a 1% (w/v) TAE agarose gel (40 mM sodium acetate, 50 mM Tris-HCl, pH 7.9 and 1 mM EDTA, pH 8.0) for 15 hr at ~2 V/cm. For visualization, gels were stained for 30 min with 0.5 μg/mL ethidium bromide in running buffer, destained in running buffer for 30 min, and exposed to UV trans-illumination. Experiments were carried out similarly for the plasmid-chase experiments, except that a 6.5 kb chase plasmid (p1C) was added with ATP to a final concentration of 10.2 μM bp when initiating reactions.

## Steady state ATP hydrolysis

ATP hydrolysis was measured using an established NADH-coupled assay (*Morrical et al., 1986*; *Tamura and Gellert, 1990*). Topo VI was thawed and diluted with 300 mM potassium glutamate, 10% (v/v) glycerol, 50 mM BTP-HCl pH 7.5 and 5 mM TCEP to 3.75 μM enzyme, mixed 1:2 with sheared salmon-sperm DNA, supercoiled pSG483, or ddH$_2$O, and incubated for 5 min on ice. Enzyme/substrate mixes were diluted with NADH-PK/LDH coupling mix to final ATP hydrolysis assay conditions [100 μL reactions, 3.75 mM phosphoenolpyruvate, 150 μM NADH, 24 U pyruvate kinase and 36 U lactate dehydrogenase (PK/LDH from rabbit muscle in buffered, aqueous glycerol solution, Sigma Aldrich, St Louis, MO), 0.1 mg/mL BSA, 50 mM BTP-HCl, pH 7.5, 50 mM potassium glutamate, 5 mM TCEP, 10 mM MgCl$_2$, 5% (v/v) glycerol, 500 nM topo VI holoenzyme]. ATP titration reactions contained either 400 μM bp sheared salmon-sperm DNA, 400 μM bp negatively supercoiled pSG483 or no DNA, and were initiated by addition of ATP to a final concentration of 0 mM or 62.5 μM-4 mM diluted in two-fold steps. DNA titrations containing 3.12–800 μM bp DNA diluted in two-fold steps were initiated by addition of ATP to a final concentration of 2 mM. Reactions were incubated at 30°C and followed in clear 96-well plates (Corning Inc, Corning, NY) by absorbance at 340 nm using a Clairiostar microplate reader. Raw absorbance values were converted to NADH molar concentrations based on measurements from NADH standards in the final ATP hydrolysis assay condition. ATP hydrolysis rates were determined by fitting to the linear portion of NADH consumption curves. Data representing three independent experiments were fit to a standard Michaelis-Menten model:

$$V_0 = \frac{k_{cat,app}[E_T][S]}{K_{m,app} + [S]} \qquad (3)$$

where $V_0$ is the observed turnover rate, $k_{cat,app}$ is the maximum turnover rate, $[E_t]$ is the total topo VI holoenzyme concentration, $[S]$ is the concentration of ATP, and $K_{m,app}$ is the Michaelis constant for ATP. For DNA titration experiments, $[S]$ is the concentration of DNA and the $k_{cat\text{-}stim,DNA}$ and $K_{stim, DNA}$ parameters substitute for $k_{cat,app}$ and $K_{m,app}$.

## Top6B dimerization assessed by FRET

Following purification, topo VI FRET constructs were labeled by reacting enzyme with 5-fold molar excess to enzyme of both Alexa Fluor 555 C$_2$ maleimide and Alexa Fluor C$_2$ 647 maleimide (ThermoFisher Scientific) in sizing buffer overnight at 4°C. TCEP was also added at 50-fold molar excess to enzyme. Reactions were quenched with 5 mM DTT and applied to a HiPrep 26/10 Desalting column (GE Healthcare Life Sciences) to separate protein from unreacted dye. Proper labeling was imaged by SDS PAGE using a Typhoon FLA 9500 laser scanner (GE Healthcare Life Sciences). Labeling efficiencies were determined by comparing absorption at 280 nm for protein to absorption at 555 nm for Alexa555 and 650 nm for Alexa647. Proteins were brought to storage buffer conditions, flash frozen as aliquots in liquid nitrogen and stored at −80°C.

For gate closure assays, labeled protein was diluted in 250 mM potassium glutamate, 10% (v/v) glycerol and 20 mM HEPES-KOH pH 7.5 to 1 µM, mixed 1:1 with 500 µM bp DNA substrate or ddH$_2$O, incubated on ice for 5 min, and diluted to final assay conditions [20 µL reactions, 200 nM topo VI, 0 or 100 µM bp DNA, 50 mM potassium glutamate, 1 mM TCEP, 10% (v/v) glycerol, 20 mM HEPES-KOH pH 7.5, 10 mM MgCl$_2$ and 0.1 mg/mL BSA]. Fluorescence emission spectra were measured by exciting samples at 530 nm and measuring emission from 545 nm to 700 nm using a Fluoromax Fluorometer 4 (HORIBA Jobin Yvon, Edison, NJ). Adenylyl-imidodiphosphate (AMPPNP) was added to a final concentration of 1 mM and changes to emission spectra were measured over time. Spectra were normalized by total emission intensity. Plotted FRET efficiencies (E) were determined ratiometrically from donor ($I_D$) and acceptor ($I_A$) peak intensities:

$$E = \frac{I_A}{I_D + I_A} \tag{4}$$

## Short DNA duplex cleavage

Topo VI was diluted in [250 mM potassium glutamate, 10% (v/v) glycerol, 10 mM MgCl$_2$ and 20 mM HEPES-KOH pH 7.5] to 1 µM, mixed 1:1 with 500 nM fluorescein-labeled duplex (*Figure 1—source data 1*) and incubated 5 min on ice. Reactions were diluted to a final cleavage reaction condition [20 µL reactions, 200 nM topo VI construct, 100 nM FAM-labeled duplex, 50 mM potassium glutamate, 1 mM TCEP, 10% (v/v) glycerol, 16 mM BTP-HCl pH 7.5, 4 mM HEPES-KOH pH 7.5, 10 mM MgCl$_2$, 0.1 mg/mL BSA and 15% DMSO]. ATP, AMPPNP or ddH$_2$O were added to initiate reactions. Reactions were incubated at 30°C for 2 hr then quenched with SDS to a final concentration of 1%. Proteinase K was added to reactions at a final concentration of 0.3 mg/mL and incubated at 45°C for 1 hr. Formamide was added 1:1 to samples and cleavage products were separated on 7 M Urea-Formamide 0.5x TBE 12% PAGE. Gels were visualized using a Typhoon FLA 9500 laser scanner.

## DNA bending assessed by FRET

DNA bending experiments used the same 70 bp duplex sequence from binding and cleavage experiments, except the substrate was modified to have a Cy5 replace the 5'-fluorocein on strand one and Cy5.5 was added to the 5' end of strand 2 (*Figure 1—source data 1*). Reactions were prepared exactly as described for the DNA cleavage assays. Fluorescence emission spectra were measured by exciting samples at 630 nm and measuring emission from 645 nm to 850 nm using a Fluoromax Fluorometer 4. AMPPNP was added to a final concentration of 1 mM and changes to emission spectra were measured over time. Spectra were normalized by total spectral emission. Plotted FRET efficiencies were calculated as for the gate closure assays.

## Expression and purification of *S. cerevisiae* topoisomerase II[1-1177] (*Sc*Top2[ΔCTR]) and *Sc*Top2[ΔCTR-cyslite-180C]

A *Sc*Top2 construct containing labeling sites on the ATP gate (*Sc*Top2[ΔCTR-cyslite-180C]) was generated from a previously described *Sc*Top2 construct with a C-terminal truncation (coding for residues 1–1177, *Sc*Top2[ΔCTR], [*Schmidt et al., 2012*]) cloned in frame with an N-terminally fused His$_6$-TEV protease-cleavable tag by introducing the following mutations: C48A, C381A, C471A, C731A. Proteins were overexpressed and purified as previously described (*Schmidt et al., 2012*). In brief, *S. cerevisiae* strain BCY123 was transformed with a GAL1 shuttle vector containing the *Sc*Top2[ΔCTR] ORF and grown in CSM-Ura$^-$ media with a 2% lactic acid and 1.5% glycerol carbon source at 30°C. Overexpression was induced by the addition of 2% galactose at A$_{600}$ = 0.8. Six hours following induction, cells were centrifuged, resuspended in 1 mM EDTA and 250 mM NaCl (1 mL/L liquid culture), and flash frozen drop-wise in liquid nitrogen.

For purification, frozen cells were first lysed under liquid nitrogen using an SPEX SamplePrep 6870 Freezer Mill (SPEX SamplePrep, Metuchen, NJ), and resultant powder was thawed and re-suspended in Buffer A300 [20 mM Tris-HCl pH 8.5, 300 mM KCl, 20 mM imidazole, and 10% (v/v) glycerol, 1 µg/mL pepstatin A, 1 µg/mL leupeptin, and 1 mM PMSF]. Lysate was clarified by centrifugation and applied to a 5 mL HiTrap Ni$^{2+}$ column equilibrated in buffer A. Following washing with buffer A, protein was eluted with buffer B [20 mM Tris-HCl pH 8.5, 100 mM KCl, 200 mM imidazole, and 10% (v/v) glycerol, 1 µg/mL pepstatin A, 1 µg/mL leupeptin, and 1 mM PMSF], and applied to a 5 mL HiTrap SP cation-exchange column. Bound protein was eluted with buffer C [20

mM Tris-HCl pH 8.5, 500 mM KCl, 10% (v/v) glycerol, 1 µg/mL pepstatin A, 1 µg/mL leupeptin and 1 mM PMSF]. Peak fractions were concentrated by centrifugation (Millipore Amicon Ultra 30K MWCO) and incubated with 1.5 mg of His$_6$ TEV protease overnight at 4°C. Uncleaved proteins and TEV protease were removed by applying the protease reaction to a HiTrap Ni$^{2+}$ column equilibrated in buffer A. Flow-through was concentrated and applied to an Sephacryl-300 HR gel filtration column (GE) equilibrated and run in ScTop2 sizing buffer [20 mM Tris-HCl pH 7.9, 500 mM KCl, 10% (v/v) glycerol]. Peak fractions were collected and concentrated (Millipore Amicon Ultra 30K MWCO). Purity was estimated by SDS-PAGE and concentration was determined by absorbance at 280 nm. ScTop2 was flash frozen in a final storage buffer containing [20 mM Tris-HCl pH 7.9, 500 mM KCl, 30% (v/v) glycerol] and stored in aliquots at −80°C.

## Supercoiled DNA relaxation by ScTop2

Plasmid relaxation assays and chase assays with ScTop2$^{\Delta CTR}$ were carried out as described for topo VI, except that ScTop2$^{\Delta CTR}$ was diluted in [500 mM KCl, 10% (v/v) glycerol, 20 mM Tris-HCl pH 7.9] and final relaxation assay conditions were [30 mM Tris-HCl pH 7.9, 10 mM MgCl$_2$, 0.05 mg/mL BSA, 0.5 mM TCEP, 100 mM KCl, 10% (v/v) glycerol, 1 mM ATP, 3.5 nM (10.2 µM bp DNA) pSG483, and 2.5 nM topo II], with 10.2 µM bp of the 6.5 kb plasmid added with ATP to initiate reactions for chase experiments.

## S. cerevisiae topo II ATPase domain dimerization assessed by FRET

The ATP gate of ScTop2$^{\Delta CTR-cyslite-180C}$ was labeled on a native cysteine residue (180C) with the Alexa Fluor 555 C$_2$ maleimide and Alexa Fluor C$_2$ 647 maleimide FRET pair following the same procedure as for Top6B, except that the reaction was carried out in ScTop2 sizing buffer and samples were flash frozen in the topo II storage buffer conditions.

Gate closure assays were performed similarly as with topo VI, except protein was diluted in 500 mM KCl, 10% (v/v) glycerol and 20 mM Tris-HCl pH 7.9, and final assay conditions were 200 nM topo II, 0 or 100 µM bp DNA, 100 mM KCl, 2% (v/v) glycerol, 10 mM Tris-HCl pH 7.9, 5 mM MgCl$_2$ and either 0 mM or 1 mM AMPPNP. Fluorescence emission spectra were measured as with topo VI.

## Data analysis and figure preparation

All data were plotted and fit using Prism Version 7 (RRID: SCR_015807, (GraphPad Software, La Jolla, CA)). Mapping of sequence conservation in relation to tertiary structure was aided by the Consurf web server (RRID: SCR_002320, [*Ashkenazy et al., 2010*]). Coordinates for bent DNA models were generated using the 3DNA web server (*Zheng et al., 2009*). Pymol was used for structure visualization and comparison (RRID: SCR_000305, [The PyMOL Molecular Graphics System, Schrödinger, LLC]).

## Acknowledgements

The authors thank current and former members of the Berger lab, as well as Scott Keeney and Corentin Claeys Bouuaert for helpful discussions and critical feedback. The authors also thank Kevin Corbett for assisting with the use of models based on previously-published SAXS data. This work was supported by a National Science Foundation graduate research fellowship (DGE 1106400 to TJW) and the National Institute of Health (RO1 CA077373 to JMB).

## Additional information

### Competing interests

James M Berger: Reviewing editor, *eLife*. The other author declares that no competing interests exist.

### Funding

| Funder | Grant reference number | Author |
| --- | --- | --- |
| National Institutes of Health | RO1 CA077373 | James M Berger |

| National Science Foundation | DGE 1106400 | Timothy J Wendorff |

The funders had no role in study design, data collection and interpretation, or the decision to submit the work for publication.

## Author contributions
Timothy J Wendorff, Conceptualization, Formal analysis, Funding acquisition, Validation, Investigation, Visualization, Methodology, Writing—original draft, Writing—review and editing; James M Berger, Conceptualization, Resources, Supervision, Funding acquisition, Visualization, Methodology, Project administration, Writing—review and editing

## Author ORCIDs
James M Berger (iD) https://orcid.org/0000-0003-0666-1240

## Decision letter and Author response
Decision letter https://doi.org/10.7554/eLife.31724.060
Author response https://doi.org/10.7554/eLife.31724.061

## Additional files

### Supplementary files
• Transparent reporting form
DOI: https://doi.org/10.7554/eLife.31724.039

### Major datasets
The following previously published datasets were used:

| Author(s) | Year | Dataset title | Dataset URL | Database, license, and accessibility information |
|---|---|---|---|---|
| Corbett KD, Benedetti P, Berger JM | 2007 | Crystal structure of the topoisomerase VI holoenzyme from Methanosarcina mazei | http://www.rcsb.org/pdb/explore.do?structureId=2q2e | Publicly available at the RCSB Protein Data Bank (accession no. 2Q2E) |
| Graille M, Cladiere L, Durand D, Lecointe F, Forterre P, van Tilbeurgh H | 2007 | Crystal structure of an intact type II DNA topoisomerase: insights into DNA transfer mechanisms | http://www.rcsb.org/pdb/explore/explore.do?structureId=2ZBK | Publicly available at the RCSB Protein Data Bank (accession no. 2ZBK) |
| Noeske J, Wasserman MR, Terry DS, Altman RB, Blanchard SC, Cate JHD | 2015 | High-resolution structure of the Escherichia coli ribosome | http://www.rcsb.org/pdb/explore/explore.do?structureId=4YBB | Publicly available at the RCSB Protein Data Bank (accession no. 4YBB) |
| Coste F, Ober M, Carell T, Boiteux S, Zelwer C, Castaing B | 2004 | Crystal Structure Complex Between the Lactococcus Lactis FPG (Mutm) and a FAPY-dG Containing DNA | http://www.rcsb.org/pdb/explore/explore.do?structureId=1TDZ | Publicly available at the RCSB Protein Data Bank (accession no. 1TDZ) |
| Guarne A, Nanji T, Gloyd M, Swiercz JP, Elliot MA | 2013 | Crystal structure of hypothetical protein SCO1480 bound to DNA | http://www.rcsb.org/pdb/explore/explore.do?structureId=4itq | Publicly available at the RCSB Protein Data Bank (accession no. 4ITQ) |
| Corbett KD, Berger JM | 2002 | Structure of topoisomerase subunit | http://www.rcsb.org/pdb/explore/explore.do?structureId=1MX0 | Publicly available at the RCSB Protein Data Bank (accession no. 1MX0) |

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

## Appendix 1

DOI: https://doi.org/10.7554/eLife.31724.040

## Supplementary material

### Comparing the expected workload of *M. mazei* topo VI to model type IIA topoisomerases

To assess whether the enzymatic activity reported for *Mm* topo VI could satisfy its cellular workload in archaea, we began by estimating the *in vitro* strand passage rate for the enzyme. Assuming an average super-helical density of $\sigma = -0.06$ for the plasmid substrate used in our assays (**Lockshon and Morris, 1983**; **Pruss et al., 1982**), each DNA molecule should contain ~17 supercoils ($Lk_o$ = 2927 bp plasmid ×1 turn/10.5 bp = 280 turns/plasmid; $\Delta Lk = \sigma \times Lk_o = -0.06 \times 280$ turns/plasmid=~17 negative supercoils/plasmid). Based on our time-course data (**Figure 1C**), at a 1:1.4 enzyme:plasmid ratio it takes between 10 and 15 min (600–900 s) for topo VI to fully relax this substrate, indicating that one strand passage event (equivalent to the removal of two supercoils) occurs every 50–75 s (i.e.: one strand passage event/2 supercoils removed ×17 supercoils/plasmid×1.4 plasmids/enzyme/600 s = 0.02 events/sec or one event every 50 s). This rate agrees well with the maximal ATPase rates we observe (~3 ATP/min on supercoiled DNA, corresponding to one strand passage event every ~40 s for an enzyme without futile cycling). Both ATPase and supercoil relaxation rates for *Mm* topo VI are at least 50-fold slower than speeds reported for a broad range of bacterial and eukaryotic type IIA topoisomerases, which catalyze strand passage events at a rate of ~1–2 $s^{-1}$ (**Basu et al., 2012**; **Higgins et al., 1978**; **Lindsley and Wang, 1993**; **Osheroff et al., 1983**; **Sugino and Cozzarelli, 1980**; **Vos et al., 2013**).

Because replication and chromosome segregation demarcate an absolute minimum requirement for topoisomerase activity in the cell, we next asked whether the genome size and doubling time of *M. mazei* under optimal growth conditions might allow for topo VI's slow activity. *M. mazei* have a ~ 8–16 hr doubling time in optimal conditions (**Mah, 1980**), and possess a ~4 Mbp genome (**Deppenmeier et al., 2002**). However, the closely related *Methanoscarina acetivorans* maintains a chromosome copy number of ~16 in similar conditions (**Hildenbrand et al., 2011**), and the maintenance of moderate to high polyploidy appears to be a common trait in euryarchaea (**Samson and Bell, 2011**). Assuming *M. mazei* possess a similar genome copy number as *M. acetivorans* during growth, each doubling would require copying ~64 Mbp of DNA per cell. *M. mazei* chromosomes are circular; thus, assuming ~10 bp/turn of DNA, replicating 64 Mbp requires the removal of ~6.4 M DNA links per cell cycle. If doubling takes 10 hr, replication in *M. mazei* requires a minimum unlinking rate of ~200 $s^{-1}$ (6.4 M DNA links/36,000 s = 178 DNA links/sec). By comparison, *E. coli* requires a similar unlinking rate to replicate and separate its single ~4 Mbp chromosome over a 40 min cell cycle (0.4M DNA links/2400 s = 167 links/sec) (**Blattner et al., 1997**; **Schaechter, 1962**). A similar minimum unlinking rate also holds for *S. cerevisiae* (12 Mbp genome/10 bp/turn of DNA/90 min doubling time yields ~220 links/sec (**Goffeau et al., 1996**; **Sherman, 2002**)). Thus, topo VI's slow activity is not compensated for by slow replication or delayed chromosome segregation when compared to bacterial and eukaryotic counterparts.

Finally, we considered whether topo VI's activity might reflect the set of encoded topoisomerases in *M. mazei,* which in addition to topo VI include topo III and a DNA gyrase gained from bacteria by horizontal gene transfer (**Forterre et al., 2007**). If *M. mazei* employs its assortment of topoisomerases similarly to other organisms, then *M. mazei* topo III would be expected to play a role in hemicatenane resolution (*Mm* topo III may provide a modest degree of negative supercoil relaxation as well, but generally this reaction is inefficient for this subfamily of type IA enzymes) (**DiGate and Marians, 1988**; **Harmon et al., 1999**; **Wallis et al., 1989**). *M. mazei* gyrase is expected to help remove positive supercoils and to further negatively-supercoil the chromosome (**Charbonnier and Forterre, 1994**; **Drlica and Snyder, 1978**; **Gellert et al., 1976**; **Higgins et al., 1978**). This division of labor suggests that *Mm* topo VI serves as the major decatenating factor of the cell and likely also assists in counterbalancing

the build-up of negative supercoils arising from transcription and gyrase activity. This role is comparable to that of topo IV in *E. coli* (*Zechiedrich and Cozzarelli, 1995*). Because the unlinking rate required by replication does not solely fall upon cellular decatenase activity (removal of both positive supercoils ahead of the replication fork and hemicatenanes also contribute (*Hiasa and Marians, 1996*; *Peter et al., 1998*)), the cellular copy number of ~5–30 topo IV heterotetramers in *E. coli* (*Taniguchi et al., 2010*; *Wiśniewski and Rakus, 2014*) may estimate the workload required of topo VI in *M. mazei*. Such a workload would require ~1500 copies of topo VI (considering 30 copies of topo IV, a topo IV strand passage rate of 1 $s^{-1}$, and a topo VI strand passage rate of 0.02 $s^{-1}$) to keep up with replication. This calculation does not begin to account for any transcriptional demands that might be also placed on topo VI, which if compared to *E. coli* would likely need to compensate for the missing activity of *E. coli* topo I in *M. mazei*, and would push the necessary copy number even higher. Collectively, these considerations indicate that topo VI either needs to be present at extremely high concentrations to fulfill its expected role in vivo, or else be significantly stimulated by as yet unidentified factors.

