## [Decision Letter]

Thank you for submitting your article "Topoisomerase VI senses and exploits both DNA crossings and bends to facilitate strand passage" for consideration by *eLife*. Your article has been reviewed by three peer reviewers, and the evaluation has been overseen by a Reviewing Editor and Andrea Musacchio as the Senior Editor. The following individuals involved in review of your submission have agreed to reveal their identity: Anthony Maxwell (Reviewer #2).

The reviewers have discussed the reviews with one another and the Reviewing Editor has drafted this decision to help you prepare a revised submission.

Summary of the work:

This paper describes mechanistic studies on a type IIB DNA topoisomerase (topo VI) from the archeon *Methanosarcina mazei*. This type of topoisomerase has some similarities to the better studied Type IIA topoisomerases, but significant differences in domain organization suggest there may be mechanistic difference between the IIA an IIB topoisomerase. The authors perform ATPase activity assays, FRET-based studies of dimerization of the N-terminal ATPase, direct and competition-based assays of topo IV binding to different types of DNA substrates (supercoiled DNA, a four-way junction or crossing, and linear DNAs of different lengths), and supercoil relaxing and processivity assays. A major part of this work is the identification of three regions of the enzyme with potential roles in DNA interaction and an exploration of these interactions using site-directed mutagenesis. Several mutants were made and tested in the same assays used with WT enzyme.

Based on these analyses, the authors conclude that topo VI senses and exploits both DNA crossings and bends to facilitate strand passage. Overall the manuscript provides new insights into how type IIB DNA topoisomerases tightly couple ATP hydrolysis and DNA cleavage to recognition of the appropriate DNA substrate. The results also have implications for understanding the mechanisms of paralogs of TopIIA such as Spo11, which helps initiate meiotic recombination.

Essential revisions:

Several issues should be clarified by the authors prior to publication.

1) Although WT topo VI binds the four-way junction (crossing or stacked junction) with a Kd of 122 nM, which is better than its binding to a 20 bp linear DNA (with a Kd of 427nM), the Kds of topo VI for 30 and 40 bp DNA are 84 and 49 nM, respectively, much better than that for the stacked junction (a total of 36 bp) (Figures 5 and 6). Why do the authors conclude that topo VI senses DNA crossing? Furthermore, does a four-way junction mimic a DNA crossing or catenate?

2) Is there any explanation for the decreased apparent affinity of topo VI for a 70 bp linear DNA compared to 40 or 60 bp DNA (Figure 1)? The minimal DNA length for topo VI binding appears to be 30 bp and the difference in Kds between 20 and 30 bp is five-fold. But why does increasing from 40 bp to 70 bp not increase DNA binding by topo VI? The increase in the Kd, app between 60 bp and 70 bp exceeds the uncertainties in the measurements. Can the authors comment on this decrease in affinity that occurs at about the length at which the model predicts that potentially both H2TH domains could interact with the duplex?

3) The ability of the H2TH region to facilitate bending a 70bp DNA substrate is strongly based on the FRET signal (shortened end-to-end distance of DNA) in the presence of H2TH but no FRET without H2TH. Is it possible that FRET arises from topo VI capturing two DNA molecules in the presence of H2TH?

4) The increased ATPase activity of the KGRR mutant compared to WT topo VI in the presence of supercoiled DNA and yet the lack of topoisomerase activity of the mutant (Figure 4) are not explained by its low affinity for supercoiled DNA and reduced ATPase dimerization, both of which are pre-requisites for ATPase activity. The supercoiled DNA-dependent ATPase domain-dimerization (FRET measurements) is initially high and decreases with time (Figure 7—figure supplement 1 and Figure 6), which is abnormal compared to the behavior of WT and the H2TH mutant topo VI, which increase with time (Figure 6B). The suggestion that the ATPase domain dissociates does not agree with the high ATPase activity, nor with the FRET data. At 150 min, the FRET signal due the ATPase domain dimerization of the KGRR mutant is at approximately the same level as the H2TH mutant topo VI and no worse. The Km of the KGRR mutant for ATP (~300µM) is much higher than WT (~3X) and H2TH topo VI (up to 10 X). KGRR is located in the ATPase domain. Could the mutation be influencing ATP binding? Do the authors know the KGRR mutant's affinity for AMPPNP (1 mM), which is used in the dimerization assay?

5) Is there any residual supercoiled DNA nicking activity of the KGRR mutant topo VI over time (Figure 4, and Figure 4—figure supplement 1)? Is there any explanation for the different "relaxed" circular DNAs in WT and three mutants? Is one a nicked band? The KGRR product is dominated by the second slowest migrating band, while the H2TH mutant predominantly produced the slowest migrating relaxed circular DNA. 10. It would be helpful if the authors could indicate linear and nicked bands on the gel for clarification. In a related point, it appears that there is an increase in the fraction of nicked or linear DNA in the gels testing the KGRR mutant in comparison to the other mutants or the wild-type enzyme. Is this significant and if so does it provide any additional insight into the regulation of cleavage by the KGRR motif?

6) It is not clear that sheared salmon sperm DNA is a suitable surrogate for linear DNA. As far as is known this preparation contains a mixture of DNAs, including single-stranded forms. Why didn't the authors just use linearized plasmid DNA in this experiment?

7) One difficult issue with the ATPase assays is the possible contribution of non-specific ATPases that may contaminate the topo VI prep. This difficulty can be addressed with other enzymes, e.g. DNA gyrase, where specific ATPase inhibitors are available. In this case the authors need to be cautious about the level of 'basal' ATPase activity, and this is particularly problematic with the mutants, where a different level of contaminating ATPase is possible. One possibility is to cut the topo VI bands out of a gel, re-fold and show that the ATPase activity is intrinsic to the enzyme. In this context it is important that the authors comment on the basal ATPase rates of the mutated enzymes.

8) Quoting k_cat_ and K_M_ values for this enzyme may not be straight forward (Figure 2—figure supplement 1). Type II topoisomerases are, in general, non-Michaelian enzymes, due to the possession of two ATPase active sites and the likely cooperativity that occurs. Although they can conform to M-M kinetics this is only 'apparent'. The ATPase data here are not really refined enough to distinguish Michaelian from non-Michaelian behaviour. However, as this is not essentially an enzyme kinetics study it does not matter too much. The authors could point this out in the text and add a footnote to the table in Figure 2—figure supplement 1 saying that these are likely to be apparent values.

9) It was surprising that the authors did not test DNA cleavage very much in this work, except in Figure 7C in the context of the mutants. Could a systematic study of cleavage couple with the binding studies have been more informative? In this context there are gels in the manuscript that show cleavage of DNA that is not commented upon, e.g. Figure 1D (large plasmid), Figure 4—figure supplement 1). Indeed the level of cleavage (linearization) is quite high in some tracks and seems to occur without ATP, which is not usually the case for topo VI. Is this a contaminant? The authors need to comment on this.

10) Subsection “Three conserved elements in Top6B play a role in DNA binding, the sensing of DNA geometry, 2 and the productive coupling of ATP hydrolysis to strand passage”, last paragraph:

The authors did the competitive binding experiments (Figure 5B) only with H2TH^EEE^ and KGRR^EEE^ but not with H2TH^AAA^ or KGRR^AAA^. Would it be possible to include the competitive binding experiments with H2TH^AAA^ and KGRR^AAA^? The authors tested these two mutants with the short-stacked junction discussed later (Figure 6B) and it would be more informative to compare the supercoil binding to these junction data for all four mutants.

11) "Together, these data indicate that both the KGRR loop and H2TH domain contribute to the preferential binding to supercoiled DNA, but that neither is solely responsible for this discrimination."

If these two domain mutations only affected supercoil DNA binding, then it could be argued that these two regions are important for supercoil DNA sensing. However, the linear DNA binding is also adversely affected for both mutations (KGRR^AAA/EEE^ and H2TH^AAA/EEE^) particularly for longer DNA (>50bp) (Figure 5—figure supplement 1). This adverse effect can be explained for H2TH since it can stabilize binding of longer DNA, but it is puzzling why KGRR^AAA^ shows lower binding affinity for longer DNA. In addition, if a positive patch is important for DNA interaction stabilized through electrostatic interactions, it is surprising that H2TH^EEE^ exhibited a lower K_d_ than that of H2TH^AAA^.

12) Subsection “The KGRR loop acts as a DNA crossing sensor to promote Top6B dimerization”, last paragraph:

In Figure 6—figure supplement 2, the authors compare the emission spectra of wild type, KGRR^AAA^ and H2TH^AAA^ with and without supercoiled DNA to probe the extent of Top6B dimerization. Interestingly, the spectra of the apo-enzymes KGRR and H2TH differ from that of the WT enzyme. For KGRR, the emission at 670 nm is higher than that of wild type while that of H2TH is lower. Is it possible that the site-specific mutation can lead to change in protein conformation? For example, two Top6B are closer or further apart for KGRR and H2TH respectively. In line with the previous comment, is it possible that this conformational change in KGRR could result in reduction in binding and or sensing of both linear DNA (G-segment) as well as the crossing DNA (T-segment)?

13) Subsection “The H2TH interface engages an extended G-segment to couple nucleotide-dependent Top6B dimerization with DNA cleavage”, second paragraph:

Regarding Figure 7: Based on the image shown in Figure 7C for the wild type case, the amounts of DNA loaded on the gel for different length of DNA appears different: the 70 bp duplex lanes appear darker than those for 60 bp, suggesting more of the 70 bp product was loaded. Is this just image artifact? Can the authors quantify the relative cleavage relative to the total in each lane? This is an important point since the argument for a critical ~70 bp length requirement for cleavage hinges on this data. Do the authors think that both H2TH should interact with DNA in order to achieve DNA cleavage or would interaction with H2TH on one side suffice? In addition, considering the fact that 20-30-40 oligo bands are well separated on the ssDNA ladder lane, the cleavage product would be expected to be located somewhere between 30-40 nt (mixture of 37 and 33 due to 2-nt stagger cut) but they appear to close to 30 nt. Do the authors have any explanation? Some of the lanes show high molecular weight bands? Are they intact duplex substrate?

14) "The inability of a 40bp duplex to support cleavage, even though this DNA binds with higher affinity than a 20bp duplex and is long enough to reach both Stalk/WKxY regions, suggests G-segment DNAs must engage both H2TH regions before strand scission can be triggered."

The authors' argument is not fully convincing. As 40bp DNA contains a preferential cleavage site in the middle (Figure 1—figure supplement 1), even if Topo VI can bind one end of H2TH as shown in Figure 7B (fifth from top configuration), the cleavage product may be too low to detect due to extremely low cleavage efficiency. Thus, the 40bp cleavage data is not sufficient to draw the conclusion of two H2TH requirement. In addition, the 60bp substrate cleavage data may suggest one-H2TH interaction is sufficient.

15) Considering the complexity of this manuscript, it is recommended the authors make things clearer regarding the conditions of experiments presented in the different sections – such as ATP, AMPPNP, DNA, enzyme concentrations and if they are all same for the same type of experiments, state as such at the beginning. Otherwise a table listing the conditions of the various measurements in one place would be helpful. It would also be useful to know how the authors decided on the reaction conditions for the topo VI assays, and how these compare with the in vivo environment of *M. mazei*. In addition it is confusing that the authors use the bp scale to convolve the length and concentration of DNA instead of indicating the length of DNA and its concentration separately for supercoiled DNA. In line with this, "800bp supercoiled DNA:enzyme" in Figure 2A can be confused as 800bp length of supercoiled DNA rather than bp scaled concentration. Whereas the rational for the scaling of the amount of DNA in bp is understandable, it would be helpful to provide the length and concentration in addition to this relative measure. It would further be useful to comment on the choice of oligos used in these experiments (Figure 1—figure supplement 1); these are based on a preferred cleavage sequence for the sulfolobus enzyme. Is there evidence that the *M. mazei* enzyme has a similar sequence preference?

---

## [Author Response]

Essential revisions:Several issues should be clarified by the authors prior to publication.1) Although WT topo VI binds the four-way junction (crossing or stacked junction) with a Kd of 122 nM, which is better than its binding to a 20 bp linear DNA (with a Kd of 427nM), the Kds of topo VI for 30 and 40 bp DNA are 84 and 49 nM, respectively, much better than that for the stacked junction (a total of 36 bp) (Figures 5 and 6). Why do the authors conclude that topo VI senses DNA crossing? Furthermore, does a four-way junction mimic a DNA crossing or catenate?

The four-way junction serves as a mimic for the crossings present in plectonemic supercoiled DNA and catenated DNA. A four-way junction will fold into a stacked-X structure at the divalent cation concentrations used in our binding experiments (see (Duckett et al., 1988; Eichman, Vargason, Mooers, and Ho, 2000; McKinney, Declais, Lilley, and Ha, 2003; Ortiz-Lombardía, González, Aymamí, Azorín, and Coll, 1999). We noticed the geometry taken on by a prospective T-segment and G-segment DNA when modeled into topo VI closely mimics this stacked-X structure. We have added two panels to Figure 6 to illustrate how a stacked junction mimics this intermediate (A and B), and have revised the text in the first paragraph of the subsection “The KGRR loop acts as a DNA crossing sensor to regulatepromote Top6B dimerization” to reference these panels.

The increased affinity of topo VI for a 30 bp or 40 bp duplex may be attributed to these DNAs engaging an extended G-segment binding interface (e.g., through the Stalk/WKxY region). The Holliday junction substrate folds as a 16 bp duplex stacked on a 20 bp duplex, and should bind no better than a simple 20 bp duplex if there were no preference for topo VI to bind to a crossing. However, the affinity does increase for the junction, indicating that topo VI favorably recognizes two DNA segments. It is from the relative increase in wildtype affinity for the stacked junction, as well as the relative decrease in affinity for this substrate with the KGRR mutants, that we conclude topo VI recognizes prospective T-segment/G-segment crossings.

2) Is there any explanation for the decreased apparent affinity of topo VI for a 70 bp linear DNA compared to 40 or 60 bp DNA (Figure 1)? The minimal DNA length for topo VI binding appears to be 30 bp and the difference in Kds between 20 and 30 bp is five-fold. But why does increasing from 40 bp to 70 bp not increase DNA binding by topo VI? The increase in the Kd, app between 60 bp and 70 bp exceeds the uncertainties in the measurements. Can the authors comment on this decrease in affinity that occurs at about the length at which the model predicts that potentially both H2TH domains could interact with the duplex?

Two lines of evidence indicate that the slightly higher K_d,app_ of topo VI for a 70 bp duplex compared to a 40 or 60 bp duplex is an effect of 5’ end-labeled fluorophore position on that particular probe DNA. First, the K_d,app_ determined by fluorescence anisotropy for an 80 bp duplex is also weaker (~2x) than the K_d,app_ determined for a 60bp duplex (see Author response image 1). However, in a competitive binding experiment, the 70 bp and 80 bp duplex compete progressively better against a fluorescently labeled probe than the 60 bp duplex (see Author response image 1). We have not included these findings with the revised manuscript, but we are happy to do so as supplements to Figure 1 if the reviewers deem it necessary.

3) The ability of the H2TH region to facilitate bending a 70bp DNA substrate is strongly based on the FRET signal (shortened end-to-end distance of DNA) in the presence of H2TH but no FRET without H2TH. Is it possible that FRET arises from topo VI capturing two DNA molecules in the presence of H2TH?

The DNA bending experiments were carried out at a 2:1 molar excess of enzyme to duplex DNA. Thus, the likelihood of two 70 bp duplexes binding to the same enzyme to induce a FRET response is small. However, to more explicitly rule out the potential capture of two duplexes as a cause for changes in FRET, we have repeated the DNA bending experiment using an equimolar mixture of two 70 bp duplexes, one labeled with Cy5 and the other with Cy5.5. In this setup, any observed FRET increase must occur between two duplexes, rather than by bending of a single DNA. We observe no nucleotide-dependent increase in FRET with this experimental setup, allowing us to rule out duplex capture as the source of the FRET signal. This control has been added as Figure 7—figure supplement 1 and is referenced in the last paragraph of the subsection “The H2TH interface engages an extended G-segment to couple nucleotide-dependent Top6B dimerization with DNA cleavage.”.

4) The increased ATPase activity of the KGRR mutant compared to WT topo VI in the presence of supercoiled DNA and yet the lack of topoisomerase activity of the mutant (Figure 4) are not explained by its low affinity for supercoiled DNA and reduced ATPase dimerization, both of which are pre-requisites for ATPase activity. The supercoiled DNA-dependent ATPase domain-dimerization (FRET measurements) is initially high and decreases with time (Figure 7—figure supplement 1 and Figure 6), which is abnormal compared to the behavior of WT and the H2TH mutant topo VI, which increase with time (Figure 6B). The suggestion that the ATPase domain dissociates does not agree with the high ATPase activity, nor with the FRET data. At 150 min, the FRET signal due the ATPase domain dimerization of the KGRR mutant is at approximately the same level as the H2TH mutant topo VI and no worse. The Km of the KGRR mutant for ATP (~300µM) is much higher than WT (~3X) and H2TH topo VI (up to 10 X). KGRR is located in the ATPase domain. Could the mutation be influencing ATP binding? Do the authors know the KGRR mutant's affinity for AMPPNP (1 mM), which is used in the dimerization assay?

The increased maximal ATPase rates of the KGRR mutants in the presence of supercoiled DNA can be explained if the release of hydrolysis products by these mutants is premature. Wildtype topo VI takes ~40 s to complete each ATPase cycle. The KGRR-AAA mutant takes ~15 s, but fails to perform strand passage. This difference suggests that under normal circumstances, a delay in product release is an important component of the topo VI strand passage mechanism. In this view, the high ATPase rates of the KGRR mutants do not reflect a ‘better’ ATPase activity, but rather an uncoupling of the ATPase cycle (and specifically, product release) from a set of slow conformational changes that are necessary for strand passage.

The experiments in Figure 4B do not measure the K_m,app_ of each topo VI mutant for ATP. Rather, as noted in the manuscript and Figure 4—figure supplement 2, these experiments look at the stimulation of ATP hydrolysis as a function of DNA concentration at a single (2 mM) ATP concentration. We apologize for any confusion our discussion of this experiment might have caused, and we have modified the parameter names to K_stim,DNA_ and k_cat-stim,DNA_. The weaker K_stim,DNA_ of the KGRR mutants agrees with the lower affinity of these mutants for supercoiled DNA.

To address whether the KGRR mutations affect ATP engagement, we have measured ATP hydrolysis by KGRR^AAA^ in the presence of supercoiled DNA (400 μM basepairs DNA) as a function of ATP concentration (**see below**). This is analogous to the experiment with wildtype topo VI in Figure 2A. The K_m,app_ for ATP is ~1.5x higher for the KGRR-AAA mutant (350 μM) than for wildtype topo VI (240 μM). This result indicates that there is a slight decrease in affinity of the KGRR-AAA mutant for ATP, but also suggests that 1 mM AMPPNP is still an appropriate concentration to use for the dimerization assay.

**Author response image 2. respfig2:** 

In the dimerization assay, the KGRR^AAA^ construct responds to AMPPNP in the presence of a 70 bp duplex, however, this response does not exceed the FRET signal generated by supercoiled DNA alone (Figure 7—figure supplement 1). While the KGRR^AAA^ construct does not respond to AMPPNP in the presence of supercoiled DNA, it does show an increased FRET signal with supercoiled DNA alone that is innately higher than wildtype topo VI or the H2TH-AAA mutant (Figure 6—figure supplement 2). These data, together with the ATPase results, suggest that upon binding supercoiled DNA, the KGRR-AAA mutant enzymes ‘pre-dimerize’ into a state competent for binding and hydrolyzing ATP (and that the intact KGRR loop of wild-type topo VI prevents this conformational change from occurring until ATP also binds). If the KGRR-AAA mutant inappropriately releases hydrolysis products before strand passage (perhaps because a bound T-segment can no longer suppress the release of these products), the enzyme will remain pre-dimerized. Interestingly, Klostermeier and colleagues have noted that ATPase pre-dimerization occurs for bacterial gyrase upon DNA binding, albeit in the wild-type enzyme (Gubaev and Klostermeier, 2011).

Finally, as noted by the reviewers, both H2TH-AAA and KGRR-AAA produce similar FRET signals in the presence of supercoiled DNA and AMPPNP that are lower than the wildtype signal in the same condition. Given both mutants are impaired for strand passage, this raises the possibility that the higher wildtype signal reflects a conformational response to nucleotide binding that is not fully accessible to either mutant. Such a model directly relates to comments raised by the reviewers in point 12 (below). We have revised the text in the last paragraph of the subsection “The KGRR loop acts as a DNA crossing sensor to regulate Top6B dimerization” to convey this interpretation more clearly.

5) Is there any residual supercoiled DNA nicking activity of the KGRR mutant topo VI over time (Figure 4, and Figure 4—figure supplement 1)? Is there any explanation for the different "relaxed" circular DNAs in WT and three mutants? Is one a nicked band? The KGRR product is dominated by the second slowest migrating band, while the H2TH mutant predominantly produced the slowest migrating relaxed circular DNA. 10. It would be helpful if the authors could indicate linear and nicked bands on the gel for clarification. In a related point, it appears that there is an increase in the fraction of nicked or linear DNA in the gels testing the KGRR mutant in comparison to the other mutants or the wild-type enzyme. Is this significant and if so does it provide any additional insight into the regulation of cleavage by the KGRR motif?

We apologize for any confusion over the identity of DNA species in Figure 4A and Figure 4—figure supplement 1. We have more clearly annotated the position of linear and nicked plasmid DNA species for these experiments. The slowest-migrating band seen for a number of mutants in Figure 4A is open-circle/nicked DNA present in the plasmid preparation. Of all the constructs, only wildtype topo VI produces a fully relaxed, closed-circle topoisomer distribution.

We do not believe excessive nicking to be a feature of the KGRR mutants. In Figure 4A, there is no dose-dependent nicking for KGRR-EEE. Any apparent dose-dependent nicking for KGRR-AAA is an artifact of gel illumination due to our old scanner; a replicated gel on a newer system with more uniform illumination displays no dose-dependent nicking (see Author response image 3).

**Author response image 3. respfig3:** 

In Figure 4—figure supplement 1, all experiments show a slowly-migrating band that increases with time. This is the open-circle/nicked DNA species. The reviewers note an additional slower-running band in the KGRR time-courses. This is contaminating plasmid concatemer in the specific plasmid preparation used in those experiments. We have replicated the time-course gels for both KGRR mutants (see Author response image 4), which verify the band that increases over time is the same nicked species that increases in the Stalk/WKxY and H2TH mutant experiments.

The KGRR timecourse replicates also show that increased nicking over time is both nucleotide and enzyme-independent. This increase may reflect either a contaminant in an assay component, or a chemical nicking activity of our 1x assay conditions over the 2 h incubation at 30 ^o^C. We have explicitly commented on this point in the manuscript (subsection “Three conserved elements in Top6B play a role in DNA binding, the sensing of DNA geometry, and the productive coupling of ATP hydrolysis to strand passage”, fourth paragraph).

**Author response image 4. respfig4:** 

6) It is not clear that sheared salmon sperm DNA is a suitable surrogate for linear DNA. As far as is known this preparation contains a mixture of DNAs, including single-stranded forms. Why didn't the authors just use linearized plasmid DNA in this experiment?

Other groups in the topoisomerase field have found sheared salmon-sperm DNA to be a suitable general dsDNA substrate (e.g., Wang, Lindsley, Nitiss and Austin groups (Baird, Gordon, Andrenyak, Marecek, and Lindsley, 2001; Harkins and Lindsley, 1998; Morris, Harkins, Tennyson, and Lindsley, 1999; Olland and Wang, 1999; Vaughn et al., 2005; Walker et al., 2004; West and Austin, 1999)). Following this precedent, we have frequently used sheared salmon-sperm DNA in previous studies and found no issue with the substrate (e.g., (Schmidt, Osheroff, and Berger, 2012; Schoeffler, May, and Berger, 2010; Tretter and Berger, 2012)). Based on agarose gel electrophoresis, the sheared salmon-sperm DNA used in this study consists of DNAs ranging from ~75-500 basepairs in length (primarily ~75-100 basepairs, see Author response image 5). This DNA mixture competes for DNA binding to topo VI similarly to a defined DNA substrate in the same length range (60 bp), whereas single stranded DNA competes quite poorly (see Author response image 5). According to the manufacturer, the sheared salmon-sperm DNA has not been boiled and should therefore be double-stranded. We confirmed that our sheared salmon-sperm DNA preparations are resistant to S1 single-stranded DNA nuclease treatment, but that when boiled, it is degraded, confirming the manufacturer’s claims (see Author response image 5). While we do not believe these controls would add to the manuscript, we are happy to include them as supplements to Figure 1 if the reviewers deem it necessary.

**Author response image 5. respfig5:** 

The DNAs in the sheared salmon-sperm sample are shorter than the persistence length of DNA and thus less likely to form bends or crossings. This characteristic makes sheared salmon-sperm DNA a good substrate to compare against supercoiled plasmid DNA, which would have such features. Because linearized plasmid DNA (~3 kb) is sufficiently long to form random bends and intramolecular crossings, we expect it to have an intermediate effect upon topo VI compared to sheared salmon sperm DNA. Two tests confirm this prediction. In the first, we have found that linearized plasmid competes better than sheared salmon-sperm DNA for topo VI binding, nearly as well as supercoiled DNA (see Author response image 6). This result comports with our model in that topo VI preferentially binds better to crossovers than single duplexes; in the absence of ATP (which is omitted in the binding assays), the crossovers present in the supercoiled and linearized plasmids will not be resolved, and will act as a sink to pull topo VI (which is present in limiting concentrations) away from the labeled 70 bp duplex. In the second test, we found that linearized plasmid has an overall stimulatory effect on ATP hydrolysis that is intermediate compared to that measured for sheared salmon-sperm DNA and supercoiled DNA (see Author response image 6). This result is interesting, because at low supercoiled plasmid:enzyme ratios, where the crossover density is low (~1 crossover/enzyme), the ATPase stimulation is similar to that of linearized plasmid. However, at higher supercoiled plasmid:enzyme ratios, where the crossover density increases, the ATPase rate improves concordantly. Overall, these data are consistent with the conclusion that topo VI senses and responds to crossings present in DNA substrates. We also have not added these data to the text, but can do so if the reviewers feel it necessary.

**Author response image 6. respfig6:** 

7) One difficult issue with the ATPase assays is the possible contribution of non-specific ATPases that may contaminate the topo VI prep. This difficulty can be addressed with other enzymes, e.g. DNA gyrase, where specific ATPase inhibitors are available. In this case the authors need to be cautious about the level of 'basal' ATPase activity, and this is particularly problematic with the mutants, where a different level of contaminating ATPase is possible. One possibility is to cut the topo VI bands out of a gel, re-fold and show that the ATPase activity is intrinsic to the enzyme. In this context it is important that the authors comment on the basal ATPase rates of the mutated enzymes.

We appreciate the potential for contamination by non-specific ATPases may contribute to the overall ATPase rates reported. We have added a supplemental figure (Figure 4—figure supplement 2) reporting the ‘basal’ ATPase rates for all mutants as a function of ATP concentration (see Figure 4—figure supplement 2). All mutants exhibit basal ATPase rates similar to each other and to wild-type topo VI.

To determine whether co-contaminants in our topo VI preparations contribute to measured ATP hydrolysis rates, we cloned, expressed, and purified a construct of topo VI that lacks a catalytic glutamate necessary for ATPase activity in GHKL ATPases (Top6B^E44A^). As expected, this mutant shows no supercoil relaxation activity (see Figure 2—figure supplement 1B,top). While there is some background ATPase activity in the Top6AB^E44A^ prep, it is not DNA stimulated (see Figure 2—figure supplement 1B, bottom). Thus, the basal rates reported for topo VI do appear to contain a contribution from a non-specific ATPase contaminant. By contrast, the ability of DNA to stimulate (or fail to stimulate) the ATPase activity of our topo VI mutants is directly attributable to the specific mutations we have introduced. To more accurately represent the topo VI-specific ATPase activity reported in Figure 2, we have subtracted out the background ‘basal’ ATPase rate measured for Top6B^E44A^, and reprocessed the kinetic parameters reported in Figure 2—figure supplement 1. This adjustment is not necessary for the experiments reported in Figure 4B, as they measure stimulation above the basal rate. We have added the experiments in panel A and B as a supplement to Figure 2 and 4. We have also added callouts to each supplement in the first paragraph of the subsection “Topo VI actively uses DNA crossings to couple ATP hydrolysis with DNA strand passage” and in “Three conserved elements in Top6B play a role in DNA binding, the sensing of DNA geometry, and the productive coupling of ATP hydrolysis to strand passage”.

8) Quoting k_cat_ and K_M_ values for this enzyme may not be straight forward (Figure 2—figure supplement 1). Type II topoisomerases are, in general, non-Michaelian enzymes, due to the possession of two ATPase active sites and the likely cooperativity that occurs. Although they can conform to M-M kinetics this is only 'apparent'. The ATPase data here are not really refined enough to distinguish Michaelian from non-Michaelian behaviour. However, as this is not essentially an enzyme kinetics study it does not matter too much. The authors could point this out in the text and add a footnote to the table in Figure 2—figure supplement 1 saying that these are likely to be apparent values.

We agree that ATP hydrolysis by topo VI is likely cooperative, but the reported data do not sufficiently assess turnover at low ATP concentrations to distinguish between a cooperative model and ‘apparent’ Michaelis-Menten behavior. We have modified both the text (subsection “Topo VI actively uses DNA crossings to couple ATP hydrolysis with DNA strand passage”, first paragraph) and Figure 2—figure supplement 1 to explicitly state that the reported kinetic parameters are apparent, and are not meant to imply that the enzyme acts in a truly Michaelian regime.

9) It was surprising that the authors did not test DNA cleavage very much in this work, except in Figure 7C in the context of the mutants. Could a systematic study of cleavage couple with the binding studies have been more informative? In this context there are gels in the manuscript that show cleavage of DNA that is not commented upon, e.g. Figure 1D (large plasmid), Figure 4—figure supplement 1). Indeed the level of cleavage (linearization) is quite high in some tracks and seems to occur without ATP, which is not usually the case for topo VI. Is this a contaminant? The authors need to comment on this.

Our goal in this work was to determine how topo VI engages DNA and how DNA binding may coordinate activities related to strand passage. The experiments performed in Figure 7C directly contribute to resolving these questions by determining both the DNA binding elements and minimal DNA length required for topo VI-mediated DNA cleavage. Given all the mutants under investigation, a truly systematic study of cleavage activities would greatly enlarge this study: DNA topology, oligo lengths, and ATP dependence would all need to be assessed for seven different constructs (eight when including our new ATPase-defective mutant to look at the role of hydrolysis). Then there is also the question of preferred cleavage sequences, both within the central cleavage region and more peripherally toward the H2TH domains. We agree that it would be useful to more systematically investigate DNA cleavage by topo VI, but as part of a future effort.

Regarding the DNA gels noted by the reviewers – in Figure 1D, topo VI produces a relaxed distribution for the large plasmid (13.5 kb) that runs very close to the nicked plasmid species. Since it is difficult to unequivocally distinguish this distribution from open-circle (nicked) plasmid, we repeated the experiment using a smaller chase plasmid (6.5 kb). There is better separation between relaxed and nicked bands for the 6.5 kb chase plasmid, which verifies topo VI does not produce a level of nicking or linearization beyond the 20 min control lacking ATP. We have substituted this experiment for Figure 1D and Figure 1—figure supplement 1.

Regarding the increase in open-circle/nicked plasmid DNA for gels in Figure 4—figure supplement 1, we have addressed this in the response to point 5, above.

10) Subsection “Three conserved elements in Top6B play a role in DNA binding, the sensing of DNA geometry, 2 and the productive coupling of ATP hydrolysis to strand passage”, last paragraph:The authors did the competitive binding experiments (Figure 5B) only with H2TH^EEE^ and KGRR^EEE^ but not with H2TH^AAA^ or KGRR^AAA^. Would it be possible to include the competitive binding experiments with H2TH^AAA^ and KGRR^AAA^? The authors tested these two mutants with the short-stacked junction discussed later (Figure 6B) and it would be more informative to compare the supercoil binding to these junction data for all four mutants.

We have added competition assays with the H2TH-AAA and KGRR-AAA mutants to Figure 5 as requested. The KGRR-AAA mutant shows intermediate effects between wildtype and the KGRR-EEE mutant. The H2TH-AAA mutant shows minimal impairment for both sheared salmon-sperm DNA and supercoiled DNA binding when compared to the H2TH-EEE mutant. We have revised the text in the subsection “Three conserved elements in Top6B play a role in DNA binding, the sensing of DNA geometry, and the productive coupling of ATP hydrolysis to strand passage”.

11) "Together, these data indicate that both the KGRR loop and H2TH domain contribute to the preferential binding to supercoiled DNA, but that neither is solely responsible for this discrimination."If these two domain mutations only affected supercoil DNA binding, then it could be argued that these two regions are important for supercoil DNA sensing. However, the linear DNA binding is also adversely affected for both mutations (KGRR^AAA/EEE^ and H2TH^AAA/EEE^) particularly for longer DNA (>50bp) (Figure 5—figure supplement 1). This adverse effect can be explained for H2TH since it can stabilize binding of longer DNA, but it is puzzling why KGRR^AAA^ shows lower binding affinity for longer DNA. In addition, if a positive patch is important for DNA interaction stabilized through electrostatic interactions, it is surprising that H2TH^EEE^ exhibited a lower K_d_ than that of H2TH^AAA^.

The ability of a site to sense supercoiled DNA does not necessarily preclude it from also impacting the binding of linear DNAs. As pointed out, the impact of the H2TH domain on linear DNA binding can be explained by the interaction of this element with longer DNA segments. As for the KGRR mutants, it may be that these substitutions affect the conformational disposition of subunits in the tetramer, which in turn makes it more difficult for DNA bound in the active site of a Top6A dimer to access the flanking Top6B WKxY and H2TH regions. The discussion outlined for points 4 and 12 comports with this reasoning.

Regarding the K_d_ values of the 60 and 70 bp DNAs for the H2TH mutants, these point mutations lie at the outer edge of the G-segment binding tract and may therefore differentially affect either DNA binding register (discussed further in point 13) and/or the mobility of the fluorophore label. To control for such effects, we have measured the affinity of both H2TH-AAA and H2TH-EEE for unlabeled 60 or 70 bp duplexes in a competition experiment that uses the fluorescently labeled 70 bp duplex as a probe. As can be seen from the data, the H2TH-EEE mutant does not exhibit greater affinity for either DNA compared to the H2TH-AAA mutant. We have added these experiments as a supplement to Figure 5.

Interestingly, whereas a 70 bp duplex competes better than a 60 bp duplex for binding to wildtype topo VI (see point 2), competitive binding of the 70 bp duplex is relatively impaired for both H2TH mutants. As we now note in the revised manuscript, this result suggests that each set of substitutions is sufficient to ablate DNA-H2TH interactions, but that they minimally affect DNA engagement by Top6A or the Stalk/WKxY region. We also note that in the competitive binding experiments reported in the new Figure 5, the H2TH-AAA mutant binds supercoiled DNA more strongly than the H2TH-EEE mutant. This finding provides additional evidence that the H2TH site engages DNA primarily in a role as a bend sensor. These considerations are explicitly discussed in the subsection “Three conserved elements in Top6B play a role in DNA binding, the sensing of DNA geometry, and the productive coupling of ATP hydrolysis to strand passage”.

12) Subsection “The KGRR loop acts as a DNA crossing sensor to promote Top6B dimerization”, last paragraph:In Figure 6—figure supplement 2, the authors compare the emission spectra of wild type, KGRR^AAA^ and H2TH^AAA^ with and without supercoiled DNA to probe the extent of Top6B dimerization. Interestingly, the spectra of the apo-enzymes KGRR and H2TH differ from that of the WT enzyme. For KGRR, the emission at 670 nm is higher than that of wild type while that of H2TH is lower. Is it possible that the site-specific mutation can lead to change in protein conformation? For example, two Top6B are closer or further apart for KGRR and H2TH respectively. In line with the previous comment, is it possible that this conformational change in KGRR could result in reduction in binding and or sensing of both linear DNA (G-segment) as well as the crossing DNA (T-segment)?

Based on the FRET spectra of the mutant enzymes alone, we agree that these point mutations could alter the underlying conformational states sampled by the enzyme, and that any such behavior might well contribute to the effect of each set of mutations on DNA binding (a concept also noted in response to points 4 and 11, above). We have revised the manuscript (subsection “The KGRR loop acts as a DNA crossing sensor to regulate Top6B dimerization”) to more explicitly comment on this possibility.

13) Subsection “The H2TH interface engages an extended G-segment to couple nucleotide-dependent Top6B dimerization with DNA cleavage”, second paragraph:Regarding Figure 7: Based on the image shown in Figure 7C for the wild type case, the amounts of DNA loaded on the gel for different length of DNA appears different: the 70 bp duplex lanes appear darker than those for 60 bp, suggesting more of the 70 bp product was loaded. Is this just image artifact? Can the authors quantify the relative cleavage relative to the total in each lane? This is an important point since the argument for a critical ~70 bp length requirement for cleavage hinges on this data. Do the authors think that both H2TH should interact with DNA in order to achieve DNA cleavage or would interaction with H2TH on one side suffice? In addition, considering the fact that 20-30-40 oligo bands are well separated on the ssDNA ladder lane, the cleavage product would be expected to be located somewhere between 30-40 nt (mixture of 37 and 33 due to 2-nt stagger cut) but they appear to close to 30 nt. Do the authors have any explanation? Some of the lanes show high molecular weight bands? Are they intact duplex substrate?

The apparent loading difference is an artifact of our previous gel scanner. A repeat of the experiment shows that loading is constant from lane to lane and that the cleavage product is still more pronounced for the 70bp oligo. We have also determined that the high molecular weight band in some lanes in Figure 7C (e.g., lanes containing wildtype topo VI and 70 bp DNA substrate) represent DNA bound to incompletely digested topo VI. These bands disappear with longer Proteinase K treatment. We have substituted these gels and quantified the% cleavage product above each lane where this product is distinguishable. Full digestion reveals that the wildtype enzyme cleaves the 70 bp DNA at a slightly higher level than either KGRR mutant. This result is concordant with the relative affinities of each construct for the 70 bp DNA. We have modified the text (subsection “The H2TH interface engages an extended G-segment to couple nucleotide-dependent Top6B dimerization with DNA cleavage”, third paragraph) to reflect these quantification data.

We agree that the cleavage bands are running at a position that would suggest topo VI is cutting off-center. Our oligo sequence is based on an optimal cleavage site identified by the Forterre group for *S. shibatae* topo VI (Buhler, Lebbink, Bocs, Ladenstein, and Forterre, 2001). The same group also identified a weaker cleavage site ~6 bp away from this locus; we suspect our *M. mazei* enzyme is cutting at this secondary site. We also agree that the data are consistent with the idea that the engagement of a single H2TH domain can be sufficient to promote cleavage. We have revised the text to reflect both points in the subsection “The H2TH interface engages an extended G-segment to couple nucleotide-dependent Top6B dimerization with DNA cleavage”.

14) "The inability of a 40bp duplex to support cleavage, even though this DNA binds with higher affinity than a 20bp duplex and is long enough to reach both Stalk/WKxY regions, suggests G-segment DNAs must engage both H2TH regions before strand scission can be triggered."The authors' argument is not fully convincing. As 40bp DNA contains a preferential cleavage site in the middle (Figure 1—figure supplement 1), even if Topo VI can bind one end of H2TH as shown in Figure 7B (fifth from top configuration), the cleavage product may be too low to detect due to extremely low cleavage efficiency. Thus, the 40bp cleavage data is not sufficient to draw the conclusion of two H2TH requirement. In addition, the 60bp substrate cleavage data may suggest one-H2TH interaction is sufficient.

We agree with this assessment of our data, and we have revised the text (subsection “The H2TH interface engages an extended G-segment to couple nucleotide-dependent Top6B dimerization with DNA cleavage”) accordingly.

15) Considering the complexity of this manuscript, it is recommended the authors make things clearer regarding the conditions of experiments presented in the different sections – such as ATP, AMPPNP, DNA, enzyme concentrations and if they are all same for the same type of experiments, state as such at the beginning. Otherwise a table listing the conditions of the various measurements in one place would be helpful. It would also be useful to know how the authors decided on the reaction conditions for the topo VI assays, and how these compare with the in vivo environment of M. mazei. In addition it is confusing that the authors use the bp scale to convolve the length and concentration of DNA instead of indicating the length of DNA and its concentration separately for supercoiled DNA. In line with this, "800bp supercoiled DNA:enzyme" in Figure 2A can be confused as 800bp length of supercoiled DNA rather than bp scaled concentration. Whereas the rational for the scaling of the amount of DNA in bp is understandable, it would be helpful to provide the length and concentration in addition to this relative measure. It would further be useful to comment on the choice of oligos used in these experiments (Figure 1—figure supplement 1); these are based on a preferred cleavage sequence for the sulfolobus enzyme. Is there evidence that the M. mazei enzyme has a similar sequence preference?

The reaction conditions (salts, buffers, etc.) for each type of experiment are the same throughout the manuscript. The only exception is between the ATP hydrolysis experiments in Figure 2 (where DNA concentration is maintained and ATP concentration is varied) and the ATP hydrolysis experiments in Figure 4B (where ATP concentration is maintained and DNA concentration is varied). As requested, we have added a table listing the enzyme concentration, nucleotide concentration and identity (i.e. ATP vs. AMPPNP), and DNA concentration for each set of experiments to aid the reader.

Assay conditions for *M. mazei* topo VI were previously optimized by screening monovalent salt identity, salt concentration, and buffer identity from a panel of common biochemical reagents (Corbett, Benedetti, and Berger, 2007). The current study uses essentially the same conditions, except for the addition of 10% (v/v) glycerol and 0.1 mg/mL BSA, which we found improved *M. mazei* topo VI activity at low enzyme concentrations. Our in vitro conditions are comparable to the optimal growth conditions and known physiology of *M. mazei,* including temperature (30 ^o^C to 40 ^o^C), pH (pH 6-8), and salinity (*M. mazei* mayharbor potassium glutamate concentrations up to 400 mM) (Deppenmeier et al., 2002; Mah, 1980; Spanheimer and Müller, 2008).

A 2.9 kb plasmid, pSG483, is used throughout the text as a supercoiled substrate. We have added this size identification to figures and text where it is missing. In cases where multiple DNA substrates are being used (i.e., ATPase assays, competition assays and gate closure assays), we have used a μM basepair DNA concentration scale. We believe this is the most straightforward scale for comparing between DNA substrates and DNA-dependent characteristics of topo VI.

The short duplexes used in this study (Figure 1—figure supplement 1) are based on a preferred sequence for *S. shibatae* topo VI. We have not explicitly tested whether *M. mazei* has a similar preference. Nevertheless, we reasoned that the *S. shibatae* topo VI cleavage sequence may contain general features recognized by type IIB topoisomerases, and using such a sequence was preferable to picking at random. We have modified the text (subsection “Topo VI is a distributive DNA relaxase that preferentially recognizes DNA crossings”, first paragraph) to highlight this point. Interestingly, in pilot experiments for cleavage of short duplexes (not discussed in the manuscript), *M. mazei* topo VI produced a small but detectable amount of cleavage of the 60 bp duplex bearing the *S. shibatae* sequence, but it did not cleave a 60 bp duplex with an unrelated sequence. We did not repeat this experiment with a 70bp oligo of random sequence.